# CONSERVATIVE SAFETY CRITICS FOR EXPLORATION

**Homanga Bharadhwaj**[1],[*]**Aviral Kumar**[2]**, Nicholas Rhinehart**[2]**,**
**Sergey Levine**[2]**, Florian Shkurti**[1]**, Animesh Garg**[1]
[1]University of Toronto, Vector Institute
[2]University of California Berkeley
homanga@cs.toronto.edu

## ABSTRACT

Safe exploration presents a major challenge in reinforcement learning (RL): when active data collection requires deploying partially trained policies, we must ensure that these policies avoid catastrophically unsafe regions, while still enabling trial and error learning. In this paper, we target the problem of safe exploration in RL by learning a conservative safety estimate of environment states through a critic, and provably upper bound the likelihood of catastrophic failures at every training iteration. We theoretically characterize the tradeoff between safety and policy improvement, show that the safety constraints are likely to be satisfied with high probability during training, derive provable convergence bounds for our approach, which is no worse asymptotically than standard RL, and demonstrate the efficacy of the proposed approach on a suite of challenging navigation, manipulation, and locomotion tasks. Empirically, we show that the proposed approach can achieve competitive task performance while incurring significantly lower catastrophic failure rates during training than prior methods. Videos are at this url https://sites.google.com/view/conservative-safety-critics/

## 1 INTRODUCTION

Reinforcement learning (RL) is a powerful framework for learning-based control because it can enable agents to learn to make decisions automatically through trial and error. However, in the real world, the cost of those trials – and those errors – can be quite high: a quadruped learning to run as fast as possible, might fall down and crash, and then be unable to attempt further trials due to extensive physical damage. However, learning complex skills without any failures at all is likely impossible. Even humans and animals regularly experience failure, but quickly learn from their mistakes and behave *cautiously* in risky situations. In this paper, our goal is to develop safe exploration methods for RL that similarly exhibit *conservative* behavior, erring on the side of caution in particularly dangerous settings, and limiting the number of catastrophic failures.

A number of previous approaches have tackled this problem of safe exploration, often by formulating the problem as a constrained Markov decision process (CMDP) (García & Fernández, 2015; Altman, 1999). However, most of these approaches require additional assumptions, like assuming access to a function that can be queried to check if a state is safe (Thananjeyan et al., 2020), assuming access to a default safe controller (Koller et al., 2018; Berkenkamp et al., 2017), assuming knowledge of all the unsafe states (Fisac et al., 2019), and only obtaining safe policies after training converges, while being unsafe during the training process (Tessler et al., 2018; Dalal et al., 2018).

In this paper, we propose a general safe RL algorithm, with bounds on the probability of failures during training. Our method only assumes access to a sparse (e.g., binary) indicator for *catastrophic failure*, in the standard RL setting. We train a *conservative* safety critic that overestimates the probability of catastrophic failure, building on tools in the recently proposed conservative Q-learning framework (Kumar et al., 2020) for offline RL. In order to bound the likelihood of catastrophic failures at every iteration, we impose a KL-divergence constraint on successive policy updates so that the stationary distribution of states induced by the old and the new policies are not arbitrarily

---

[*]Work done during HB's (virtual) visit to Sergey Levine's lab at UC Berkeley

different. Based on the safety critic's value, we consider a chance constraint denoting probability of failure, and optimize the policy through primal-dual gradient descent.

Our key contributions in this paper are designing an algorithm that we refer to as *Conservative Safety Critics* (CSC), that learns a conservative estimate of how safe a state is, using this conservative estimate for safe-exploration and policy updates, and theoretically providing upper bounds on the probability of failures throughout training. Through empirical evaluation in five separate simulated robotic control domains spanning manipulation, navigation, and locomotion, we show that CSC is able to learn effective policies while reducing the rate of catastrophic failures by up to 50% over prior safe exploration methods.

## 2 PRELIMINARIES

We describe the problem setting of a constrained MDP (Altman, 1999) specific to our approach and the conservative Q learning (Kumar et al., 2020) framework that we build on in our algorithm.

**Constrained MDPs.** We take a constrained RL view of safety (García & Fernández, 2015; Achiam et al., 2017), and define safe exploration as the process of ensuring the constraints of the constrained MDP (CMDP) are satisfied while exploring the environment to collect data samples. A CMDP is a tuple $(\mathcal{S}, \mathcal{A}, P, R, \gamma, \mu, \mathcal{C})$, where $\mathcal{S}$ is the state space, $\mathcal{A}$ is the action space, $P : \mathcal{S} \times \mathcal{A} \times \mathcal{S} \to [0, 1]$ is a transition kernel, $R : \mathcal{S} \times \mathcal{A} \to \mathbb{R}$ is a task reward function, $\gamma \in (0, 1)$ is a discount factor, $\mu$ is a starting state distribution, and $\mathcal{C} = \{(c_i : \mathcal{S} \to \{0, 1\}, \chi_i \in \mathbb{R}) | i \in \mathbb{Z}\}$ is a set of (safety) constraints that the agent must satisfy, with constraint functions $c_i$ taking values either 0 (*alive*) or 1 (*failure*) and limits $\chi_i$ defining the maximal allowable amount of non-satisfaction, in terms of expected probability of failure. A stochastic policy $\pi : \mathcal{S} \to \mathcal{P}(\mathcal{A})$ is a mapping from states to action distributions, and the set of all stationary policies is denoted by $\Pi$. Without loss of generality, we can consider a *single* constraint, where $\mathcal{C}$ denotes the constraint satisfaction function $C : \mathcal{S} \to \{0, 1\}$, $(C \equiv \mathbb{1}\{failure\})$ similar to the task reward function, and an upper limit $\chi$. Note that since we assume only a sparse binary indicator of failure from the environment $C(s)$, in purely online training, the agent *must* fail a few times during training, and hence 0 failures is impossible. However, we will discuss how we can minimize the number of failures to a small rate, for constraint satisfaction.

We define discounted state distribution of a policy $\pi$ as $d^\pi(s) = (1 - \gamma) \sum_{t=0}^\infty \gamma^t P(s_t = s | \pi)$, the state value function as $V_R^\pi(s) = \mathbb{E}_{\tau \sim \pi}[R(\tau)|s_0 = s]$, the state-action value function as $Q_R^\pi(s, a) = \mathbb{E}_{\tau \sim \pi}[R(\tau)|s_0 = s, a_0 = a]$, and the advantage function as $A_R^\pi(s, a) = Q_R^\pi(s, a) - V_R^\pi(s)$. We define similar quantities for the constraint function, as $V_C$, $Q_C$, and $A_C$. So, we have $V_R^\pi(\mu) = \mathbb{E}_{\tau \sim \pi}[\sum_{t=0}^\infty R(s_t, a_t)]$ and $V_C^\pi(\mu)$ denoting the average episodic failures, which can also be interpreted as *expected probability of failure* since $V_C^\pi(\mu) = \mathbb{E}_{\tau \sim \pi}[\sum_{t=0}^\infty C(s_t)] = \mathbb{E}_{\tau \sim \pi}[\mathbb{1}\{failure\}] = \mathbb{P}(failure|\mu)$. For policy parameterized as $\pi_\phi$, we denote $d^\pi(s)$ as $\rho_\phi(s)$. Note that although $C : \mathcal{S} \to \{0, 1\}$ takes on binary values in our setting, the function $V_C^\pi(\mu)$ is a continuous function of the policy $\pi$.

**Conservative Q Learning.** CQL (Kumar et al., 2020) is a method for offline/batch RL (Lange et al., 2012; Levine et al., 2020) that aims to learn a $Q$-function such that the expected value of a policy under the learned $Q$ function lower bounds its true value, preventing over-estimation due to out-of-distribution actions as a result. In addition to training Q-functions via standard Bellman error, CQL minimizes the expected $Q$-values under a particular distribution of actions, $\mu(a|s)$, and maximizes the expected Q-value under the on-policy distribution, $\pi(a|s)$. CQL in and of itself might lead to unsafe exploration, whereas we will show in Section 3, how the theoretical tool introduced in CQL can be used to devise a safe RL algorithm.

## 3 THE CONSERVATIVE SAFE-EXPLORATION FRAMEWORK

In this section we describe our safe exploration framework. The safety constraint $C(s)$ defined in Section 2 is an indicator of catastrophic failure: $C(s) = 1$ when a state $s$ is unsafe and $C(s) = 0$ when it is not, and we ideally desire $C(s) = 0 \ \forall s \in \mathcal{S}$ that the agent visits. Since we do not make any assumptions in the problem structure for RL (for example a known dynamics model), we cannot guarantee this, but can at best reduce the *probability of failure* in every episode. So, we formulate the constraint as $V_C^\pi(\mu) = \mathbb{E}_{\tau \sim \pi}[\sum_{t=0}^\infty C(s_t)] \leq \chi$, where $\chi \in [0, 1)$ denotes *probability of failure*. Our approach is motivated by the insight that by being "conservative" with respect to how

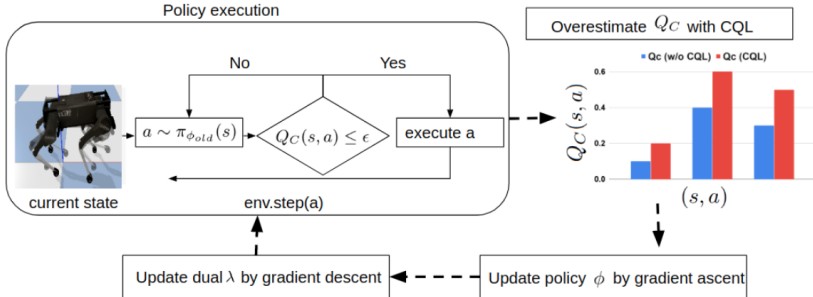

Figure 1: **CSC (Algorithm 1)**. $env.step(a)$ steps the simulator to the next state $s'$ and provides $R(s, a)$ and $C(s')$ values to the agent. If $C(s') = 1$ (*failure*), episode terminates. $Q_C$ is the learned safety critic.

safe a state is, and hence by over-estimating this probability of failure, we can effectively ensure constrained exploration.

Figure 1 provides an overview of the approach. The key idea of our algorithm is to train a conservative safety critic denoted as $Q_C(s, a)$, that overestimates how unsafe a particular state is and modifies the exploration strategy to appropriately account for this safety under-estimate (by overestimating the probability of failure). During policy evaluation in the environment, we use the safety critic $Q_C(s, a)$ to reduce the chance of catastrophic failures by checking whether taking action $a$ in state $s$ has $Q_C(s, a)$ less than a threshold $\epsilon$. If not, we re-sample $a$ from the current policy $\pi(a|s)$.

We now discuss our algorithm more formally. We start by discussing the procedure for learning the safety critic $Q_C$, then discuss how we incorporate this in the policy gradient updates, and finally discuss how we perform safe exploration (Garcıa & Fernández, 2015) during policy execution in the environment.

**Overall objective.** Our objective is to learn an optimal policy $\pi^*$ that maximizes task rewards, while respecting the constraint on expected probability of failures.

$$\pi^* = \arg\max_{\pi \in \Pi_C} V_R^\pi(\mu) \quad \text{where} \quad \Pi_C = \{\pi \in \Pi : V_C^\pi(\mu) \leq \chi\} \tag{1}$$

**Learning the safety critic.** The safety critic $Q_C$ is used to obtain an estimate of how unsafe a particular state is, by providing an estimate of *probability of failure*, that will be used to guide exploration. We desire the estimates to be "conservative", in the sense that the probability of failure should be an *over-estimate* of the actual probability so that the agent can err on the side of caution while exploring. To train such a critic $Q_C$, we incorporate tools from CQL to estimate $Q_C$ through updates similar to those obtained by reversing the sign of $\alpha$ in Equation 2 of CQL($\mathcal{H}$) (Kumar et al., 2020). This gives us an *upper bound* on $Q_C$ instead of a lower bound, as ensured by CQL. We denote the over-estimated advantage corresponding to this safety critic as $\hat{A}_C$.

Formally the safety critic is trained via the following objective, where the objective inside $\arg\min$ is called $CQL(\zeta)$, $\zeta$ parameterizes $Q_C$, and $k$ denotes the $k^{\text{th}}$ update iteration.

$$\hat{Q}_C^{k+1} \leftarrow \arg\min_{Q_C} \ \textcolor{red}{\alpha} \cdot \left(-\mathbb{E}_{s\sim\mathcal{D}_{env},a\sim\pi_\phi(a|s)}[Q_C(s,a)] + \mathbb{E}_{(s,a)\sim\mathcal{D}_{env}}[Q_C(s,a)]\right)$$
$$+ \frac{1}{2}\mathbb{E}_{(s,a,s',c)\sim\mathcal{D}_{env}}\left[\left(Q_C(s,a) - \hat{\mathcal{B}}^{\pi_\phi}\hat{Q}_C^k(s,a)\right)^2\right] \tag{2}$$

Here, $\hat{\mathcal{B}}^{\pi_\phi}$ is the empirical Bellman operator discussed in section 3.1 and equation 2 of Kumar et al. (2020). $\alpha$ is a weight that varies the importance of the first term in equation 2, and controls the magnitude of value over-estimation, as we now highlight in red above. For states sampled from the replay buffer $\mathcal{D}_{env}$, the first term seeks to maximize the expectation of $Q_C$ over actions sampled from the current policy, while the second term seeks to minimize the expectation of $Q_C$ over actions sampled from the replay buffer. $\mathcal{D}_{env}$ can include off-policy data, and also offline-data (if available). We interleave the gradient descent updates for training of $Q_C$, with gradient ascent updates for policy $\pi_\phi$ and gradient descent updates for Lagrange multiplier $\lambda$, which we describe next.

**Policy learning.** Since we want to learn policies that obey the constraint we set in terms of the safety critic, we can solve the objective in equation 1 via:

$$\max_{\pi_\phi} \ \mathbb{E}_{s\sim\rho_\phi,a\sim\pi_\phi}\left[A_R^{\pi_\phi}(s,a)\right] \quad \text{s.t.} \quad \mathbb{E}_{s\sim\rho_\phi,a\sim\pi_\phi}Q_C(s,a) \leq \chi \tag{3}$$

We can construct a Lagrangian and solve the policy optimization problem through primal dual gradient descent

$$\max_{\pi_\phi} \min_{\lambda \geq 0} \mathbb{E}_{s \sim \rho_\phi, a \sim \pi_\phi} \left[ A_R^{\pi_\phi}(s,a) - \lambda \left( Q_C(s,a) - \chi \right) \right]$$

We can apply vanilla policy gradients or some actor-critic style Q-function approximator for optimization. Here, $Q_C$ is the safety critic trained through CQL as described in equation 2. We defer specific implementation details for policy learning to the final paragraph of this section.

---

**Algorithm 1** CSC: safe exploration with conservative safety critics

---

1: Initialize $V_\theta^r$ (task value fn), $Q_\zeta^s$ (safety critic), policy $\pi_\phi$, $\lambda$, $\mathcal{D}_{env}$, thresholds $\epsilon, \delta, \chi$.
2: Set $\hat{V}_C^{\pi_{\phi_{old}}}(\mu) \leftarrow \chi$.             ▷ $\hat{V}_C^{\pi_{\phi_{old}}}(\mu)$ denotes avg. failures in the *previous* epoch.
3: **for** epochs until convergence **do**       ▷ Execute actions in the environment. Collect on-policy samples.
4:     **for** episode $e$ in $\{1, \dots, M\}$ **do**
5:         Set $\epsilon \leftarrow (1-\gamma)(\chi - \hat{V}_C^{\pi_{\phi_{old}}}(\mu))$
6:         Sample $a \sim \pi_{\phi_{old}}(s)$. Execute $a$ iff $Q_C(s,a) \leq \epsilon$. Else, resample $a$.
7:         Obtain next state $s'$, $r = R(s,a)$, $c = C(s')$.
8:         $\mathcal{D}_{env} \leftarrow \mathcal{D}_{env} \cup \{(s,a,s',r,c)\}$     ▷ If available, $\mathcal{D}_{env}$ can be seeded with off-policy/offline data
9:     **end for**
10:    Store the average episodic failures $\hat{V}_C^{\pi_{\phi_{old}}}(\mu) \leftarrow \sum_{e=1}^M \hat{V}_C^e$
11:    **for** step $t$ in $\{1, \dots, N\}$ **do**           ▷ Policy and Q function updates using $\mathcal{D}_{env}$
12:         Gradient ascent on $\phi$ and (Optionally) add Entropy regularization (Appendix A.2)
13:         Gradient updates for the Q-function $\zeta := \zeta - \eta_Q \nabla_\zeta CQL(\zeta)$
14:         Gradient descent step on Lagrange multiplier $\lambda$ (Appendix A.2)
15:    **end for**
16:    $\phi_{old} \leftarrow \phi$
17: **end for**

---

**Executing rollouts (i.e., safe exploration).** Since we are interested in minimizing the number of constraint violations while exploring the environment, we do not simply execute the learned policy iterate in the environment for active data collection. Rather, we query the safety critic $Q_C$ to obtain an estimate of how unsafe an action is and choose an action that is safe via rejection sampling. Formally, we sample an action $a \sim \pi_{\phi_{old}}(s)$, and check if $Q_C(s,a) \leq \epsilon$.

We keep re-sampling actions $\pi_{\phi_{old}}(s)$ until this condition is met, and once met, we execute that action in the environment. In practice, we execute this loop for 100 iterations, and choose the action $a$ among all actions in state $s$ for which $Q_C(s,a) \leq \epsilon$ and the value of $Q_C(s,a)$ is minimum. If no such action $a$ is found that maintains $Q_C(s,a) \leq \epsilon$, we just choose $a$ for which $Q_C(s,a)$ is minimum (although above the threshold).

Here, $\epsilon$ is a threshold that varies across iterations and is defined as $\epsilon = (1-\gamma)(\chi - \hat{V}_C^{\pi_{\phi_{old}}}(\mu))$ where, $\hat{V}_C^{\pi_{\phi_{old}}}(\mu)$ is the average episodic failures in the *previous* epoch, denoting a sample estimate of the true $V_C^{\pi_{\phi_{old}}}(\mu)$. This value of $\epsilon$ is theoretically obtained such that Lemma 1 holds.

In the replay buffer $\mathcal{D}_{env}$, we store tuples of the form $(s, a, s', r, c)$, where $s$ is the previous state, $a$ is the action executed, $s'$ is the next state, $r$ is the task reward from the environment, and $c = C(s')$, the constraint value. In our setting, $c$ is binary, with 0 denoting a *live* agent and 1 denoting *failure*.

**Overall algorithm.** Our overall algorithm, shown in Algorithm 1, executes policy rollouts in the environment by respecting the constraint $Q_C(s,a) \leq \epsilon$, stores the observed data tuples in the replay buffer $\mathcal{D}_{env}$, and uses the collected tuples to train a safety value function $Q_C$ using equation 2, update the policy and the dual variable $\lambda$ following the optimization objective in equation 6.

**Implementation details.** Here, we discuss the specifics of the implementation for policy optimization. We consider the surrogate policy improvement problem Sutton (2020):

$$\max_{\pi_\phi} \quad \mathbb{E}_{s \sim \rho_{\phi_{old}}, a \sim \pi_\phi} \left[ A_R^{\pi_{\phi_{old}}}(s,a) \right] \tag{4}$$

$$\text{s.t.} \quad \mathbb{E}_{s \sim \rho_{\phi_{old}}}[D_{\text{KL}}(\pi_{\phi_{old}}(\cdot|s)||\pi_\phi(\cdot|s))] \leq \delta \quad \text{and} \quad V_C^{\pi_\phi}(\mu) \leq \chi$$

Here, we have introduced a $D_{\text{KL}}$ constraint to ensure successive policies are *close* in order to help obtain bounds on the expected failures of the new policy in terms of the expected failures of the old policy in Section 4. We replace the $D_{\text{KL}}(\pi_{\phi_{old}}(\cdot|s)||\pi_\phi(\cdot|s))$ term by its second order Taylor expansion (expressed in terms of the Fisher Information Matrix $\boldsymbol{F}$) and enforce the resulting constraint

exactly (Schulman et al., 2015a). Following equation 22 (Appendix A.2) we have,

$$\max_{\pi_\phi} \mathbb{E}_{s\sim\rho_{\phi_{old}},a\sim\pi_\phi}\left[A_R^{\pi_{\phi_{old}}}(s,a)\right] \quad \text{s.t.} \quad V_C^{\pi_{\phi_{old}}}(\mu) + \frac{1}{1-\gamma}\mathbb{E}_{s\sim\rho_{\phi_{old}},a\sim\pi_\phi}[A_C(s,a)] \leq \chi$$

$$\text{s.t.} \quad \mathbb{E}_{s\sim\rho_{\phi_{old}}}[D_{\text{KL}}(\pi_{\phi_{old}}(\cdot|s)||\pi_\phi(\cdot|s))] \leq \delta \tag{5}$$

We replace the true $A_C$ by the learned over-estimated $\hat{A}_C$, and consider the Lagrangian dual of this constrained problem, which we can solve by alternating gradient descent as shown below.

$$\max_{\pi_\phi} \min_{\lambda\geq 0} \mathbb{E}_{s\sim\rho_{\phi_{old}},a\sim\pi_\phi}\left[A_R^{\pi_{\phi_{old}}}(s,a)\right] - \lambda\left(V_C^{\pi_{\phi_{old}}}(\mu) + \frac{1}{1-\gamma}\mathbb{E}_{s\sim\rho_{\phi_{old}},a\sim\pi_\phi}\left[\hat{A}_C(s,a)\right] - \chi\right)$$

$$\text{s.t.} \quad \frac{1}{2}(\phi-\phi_{old})^T \boldsymbol{F}(\phi-\phi_{old}) \leq \delta \tag{6}$$

Note that although we use FIM for the updates, we can also apply vanilla policy gradients or some actor-critic style Q-function approximator to optimize equation 6. Detailed derivations of the gradient updates are in Appendix A.2.

## 4 THEORETICAL ANALYSIS

In this section, we aim to theoretically analyze our approach, showing that the expected probability of failures is bounded after each policy update throughout the learning process, while ensuring that the convergence rate to the optimal solution is only mildly bottlenecked by the additional safety constraint.

Our main result, stated in Theorem 1, bounds the expected probability of failure of the policy that results from Equation 5. To prove this, we first state a Lemma that shows that the constraints in Equation 5 are satisfied with high probability during the policy updates. Detailed proofs of all the Lemmas and Theorems are in Appendix A.1.

**Notation.** Let $\epsilon_C = \max_s |\mathbb{E}_{a\sim\pi_{\phi_{new}}} A_C(s,a)|$ and $\Delta$ be the amount of overestimation in the expected advantage value generated from the safety critic, $\mathbb{E}_{s\sim\rho_{\phi_{old'}},a\sim\pi_{\phi_{old}}}[\hat{A}_C(s,a)]$ as per equation 2, such that $\Delta = \mathbb{E}_{s\sim\rho_{\phi_{old'}},a\sim\pi_{\phi_{old}}}[\hat{A}_C(s,a) - A_C(s,a)]$. Let $\zeta$ denote the sampling error in the estimation of $V_C^{\pi_{\phi_{old}}}(\mu)$ by its sample estimate $\hat{V}_C^{\pi_{\phi_{old}}}(\mu)$ (i.e. $\zeta = |\hat{V}_C^{\pi_{\phi_{old}}}(\mu) - V_C^{\pi_{\phi_{old}}}(\mu)|$) and $N$ be the number of samples used in the estimation of $V_C$. Let $\text{Reg}_C(T)$ be the total cumulative failures incurred by running Algorithm 1 until $T$ samples are collected from the environment. We first show that when using Algorithm 1, we can upper bound the expectation probability of failure for each policy iterate $\pi_{\phi_{old}}$.

**Lemma 1.** *If we follow Algorithm 1, during policy updates via Equation 5, the following is satisfied with high probability $\geq 1 - \omega$*

$$V_C^{\pi_{\phi_{old}}}(\mu) + \frac{1}{1-\gamma}\mathbb{E}_{s\sim\rho_{\phi_{old}},a\sim\pi_\phi}[A_C(s,a)] \leq \chi + \zeta - \frac{\Delta}{1-\gamma}$$

*Here, $\zeta$ captures sampling error in the estimation of $V_C^{\pi_{\phi_{old}}}(\mu)$ and we have $\zeta \leq \frac{C'\sqrt{\log(1/\omega)}}{|N|}$, where $C'$ is a constant independent of $\omega$ obtained from union bounds and concentration inequalities (Kumar et al., 2020) and $N$ is the number of samples used in the estimation of $V_C$.*

This lemma intuitively implies that the constraint on the safety critic in equation 5 is satisfied with a high probability, when we note that the RHS can be made small as $N$ becomes large.

Lemma 1 had a bound in terms of $V_C^{\pi_{\phi_{old}}}(\mu)$ for the old policy $\pi_{\phi_{old}}$, but not for the updated policy $\pi_{\phi_{new}}$. We now show that the expected probability of failure for the policy $\pi_{\phi_{new}}$ resulting from solving equation 5, $V_C^{\pi_{\phi_{new}}}(\mu)$ is bounded with a high probability.

**Theorem 1.** *Consider policy updates that solve the constrained optimization problem defined in Equation 5. With high probability $\geq 1 - \omega$, we have the following upper bound on expected probability of failure $V_C^{\pi_{\phi_{new}}}(\mu)$ for $\pi_{\phi_{new}}$ during every policy update iteration:*

$$V_C^{\pi_{\phi_{new}}}(\mu) \leq \chi + \zeta - \frac{\Delta}{1-\gamma} + \frac{\sqrt{2\delta}\gamma\epsilon_C}{(1-\gamma)^2} \quad \text{where} \quad \zeta \leq \frac{C'\sqrt{\log(1/\omega)}}{|N|} \tag{7}$$

So far we have shown that, with high probability, we can satisfy the constraint in the objective during policy updates (Lemma 1) and obtain an upper bound on the expected probability of failure of the updated policy $\pi_{\phi_{new}}$ (Theorem 1). The key insight from Theorem 1 is that if we execute policy $\pi_{\phi_{new}}$ in the environment, the probability of failing is upper-bounded by a small number depending on the specified safety threshold $\chi$. Since the probability of failure is bounded, if we execute $\pi_{\phi_{new}}$ for multiple episodes, the total number of failures is bounded as well.

We now bound the task performance in terms of policy return and show that incorporating and satisfying safety constraints during learning does not severely affect the convergence rate to the optimal solution for task performance. Theorem 2 builds upon and relies on the assumptions in (Agarwal et al., 2019) and extends it to our constrained policy updates in equation 5.

**Theorem 2** (Convergence rate for policy gradient updates with the safety constraint). *If we run the policy gradient updates through equation 5, for policy $\pi_\phi$, with $\mu$ as the starting state distribution, with $\phi^{(0)} = 0$, learning rate $\eta > 0$, and choose $\alpha$ as mentioned in the discussion of Theorem 1, then for all policy update iterations $T > 0$ we have, with probability $\geq 1 - \omega$,*

$$V_R^*(\mu) - V_R^{(T)}(\mu) \leq \frac{\log |\mathcal{A}|}{\eta T} + \frac{1}{(1-\gamma)^2 T} + K \frac{\sum_{t=0}^{T-1} \lambda^{(t)}}{\eta T} \quad where \quad K \leq (1-\chi) + \frac{4\sqrt{2\delta}\gamma}{(1-\gamma)^2}$$

Since the value of the dual variables $\lambda$ strictly decreases during gradient descent updates (Algorithm 1), $\sum_{t=0}^{T-1} \lambda^{(t)}$ is upper-bounded. So, we see that the additional term proportional to $K$ introduced in the convergence rate (compared to (Agarwal et al., 2019)) due to the safety constraint is upper bounded, and can be made small with a high probability by choosing $\alpha$ appropriately. In addition, we note that the safety threshold $\chi$ helps tradeoff the convergence rate by modifying the magnitude of $K$ (a low $\chi$ means a stricter safety threshold, and a higher value of $K$, implying a larger RHS and slower convergence). We discuss some practical considerations of the theoretical results in Appendix A.4.

So far we have demonstrated that the resulting policy iterates from our algorithm all satisfy the desired safety constraint of the CMDP which allows for a maximum safety violation of $\chi$ for every intermediate policy. While this result ensures that the probability of failures is bounded, it does not elaborate on the total failures incurred by the algorithm. In our next result, we show that the cumulative failures until a certain number of samples $T$ of the algorithm grows sublinearly when executing Algorithm 1, provided the safety threshold $\chi$ is set in the right way.

**Theorem 3.** *[Number of cumulative safety failures grows sublinearly] Let $\chi$ in Algorithm 1 be time-dependent such that $\chi_t = \mathcal{O}(1/\sqrt{t})$. Then, the total number of cumulative safety violations until when $T$ transition samples have been collected by Algorithm 1, $Reg_C(T)$, scales sub-linearly with $T$, i.e., $Reg_C(T) = \mathcal{O}(\sqrt{|\mathcal{S}||\mathcal{A}|T})$.*

A proof is provided in Appendix A.1. Theorem 3 is in many ways similar to a typical regret bound for exploration (Russo, 2019; Jaksch et al., 2010), though it measures the total number of safety violations. This means that training with Algorithm 1 will converge to a "safe" policy that incurs no failures at a quick, $\mathcal{O}(\sqrt{T})$ rate.

## 5 EXPERIMENTS

Through experiments on continuous control environments of varying complexity, we aim to empirically evaluate the agreement between empirical performance and theoretical guidance by understanding the following questions:

- How *safe* is CSC in terms of constraint satisfaction during training?
- How does learning of *safe policies* trade-off with task performance during training?

### 5.1 EXPERIMENTAL SETUP

**Environments.** In each environment, shown in Figure 2, we define a task objective that the agent must achieve and a criteria for *catastrophic failure*. The goal is to solve the task without dying. In *point agent/car navigation avoiding traps*, the agent must navigate a maze while avoiding traps. The agent has a health counter that decreases every timestep that it spends within a trap. When the

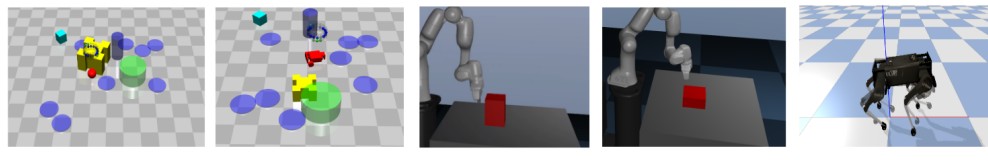

(a)  2D nav.      (b)  Car nav.      (c)  Panda topple   (d)  Panda boundary   (e)  Laikago

Figure 2: Illustrations of the five environments in our experiments: (a) 2D Point agent navigation avoiding traps. (b) Car navigation avoiding traps. (c) Panda push without toppling. (d) Panda push within boundary. (e) Laikago walk without falling.

counter hits 0, the agent gets *trapped* and *dies*. In **Panda push without toppling**, a 7-DoF Franka Emika Panda arm must push a vertically placed block across the table to a goal location without the block toppling over. *Failure* is defined as when the block topples. In **Panda push within boundary**, the Panda arm must be controlled to push a block across the table to a goal location without the block going outside a rectangular constraint region. *Failure* occurs when the block center of mass ($(x, y)$ position) move outside the constraint region. In **Laikago walk without falling**, an 18-DoF Laikago quadruped robot must walk without falling. The agent is rewarded for walking as fast as possible (or trotting) and *failure* occurs when the robot falls. Since quadruped walking is an *extremely* challenging task, for all the baselines, we initialize the agent's policy with a controller that has been trained to keep the agent standing, while not in motion.

**Baselines and comparisons.** We compare CSC to three prior methods: constrained policy optimization (**CPO**) (Achiam et al., 2017), a standard unconstrained RL method (Schulman et al., 2015a) which we call **Base** (comparison with SAC (Haarnoja et al., 2018) in Appendix Figure 7), an algorithm similar to Base, called **BaseShaped** that modifies the reward $R(s, a)$ as $R(s, a) - PC(s)$ where $P = 10$ and $C(s)$ is 1 when a failure occurs and is 0 otherwise. We also consider a method that extends Leave No Trace (Eysenbach et al., 2017) to our setting, which we refer to as **Q ensembles**. This last comparison is the most similar to our approach, in that it also implements a safety critic (adapted from LNT's backward critic), but instead of using our conservative updates, the safety critic uses an ensemble for epistemic uncertainty estimation, as proposed by Eysenbach et al. (2017).

There are other safe RL approaches which we cannot compare against, as they make multiple additional assumptions, such as the availability of a function that can be queried to determine if a state is safe or not Thananjeyan et al. (2020), availability of a default safe policy for the task Koller et al. (2018); Berkenkamp et al. (2017), and prior knowledge of the location of unsafe states (Fisac et al., 2019). In addition to the baselines (Figure 3), we analyze variants of our algorithm with different safety thresholds through ablation studies (Figure 4). We also analyze CSC and the baselines by seeding with a small amount of offline data in the Appendix A.10.

## 5.2 EMPIRICAL RESULTS

**Comparable or better performance with significantly lower failures during training.** In Figure 3, we observe that CSC has significantly lower average failures per episode, and hence lower cumulative failures during the entire training process. Although the failures are significantly lower for our method, task performance and convergence of average task rewards is comparable to or better than all prior methods, including the *Base* method, corresponding to an unconstrained RL algorithm. While the *CPO* and *Q-ensembles* baselines also achieve near 0 average failures eventually, we see that CSC achieves this very early on during training.

**CSC trades off performance with safety constraint satisfaction, based on the safety-threshold $\chi$.** In Figure 4, we plot variants of our method with different safety constraint thresholds $\chi$. Observe that: (a) when the threshold is set to a lower value (stricter constraint), the number of avg. failures per episode decreases in all the environments, and (b) the convergence rate of the task reward is lower when the safety threshold is stricter. These observations empirically complement our theoretical guarantees in Theorems 1 and 2. We note that there are quite a few failures even in the case where $\chi = 0.0$, which is to be expected in practice because in the initial stages of training there is high function approximation error in the learned critic $Q_C$. However, we observe that the average episodic failures quickly drop below the specified threshold after about 500 episodes of training.

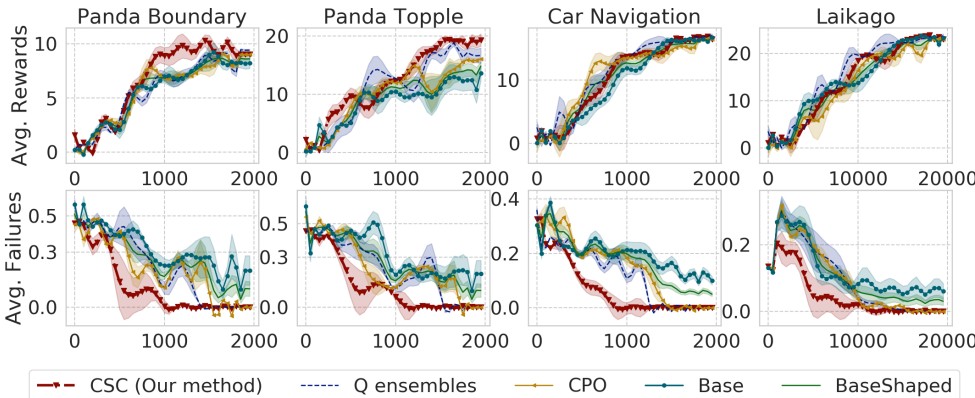

Figure 3: **Top row:** Average task rewards (higher is better). **Bottom row:** Average catastrophic failures (lower is better). **x-axis:** Number of episodes (each episode has 500 steps). Results on four of the five environments we consider for our experiments. For each environment, we plot the average task reward, the average episodic failures, and the cumulative episodic failures. The safety threshold is $\chi = 0.03$ for all the baselines in all the environments. Results are over four random seeds. Detailed results including plots of cumulative failures are in Fig. 6 of the Appendix.

## 6 RELATED WORK

We discuss prior safe RL and safe control methods under three subheadings

**Assuming prior domain knowledge of the problem structure.** Prior works have attempted to solve safe exploration in the presence of structural assumptions about the environment or safety structures. For example, Koller et al. (2018); Berkenkamp et al. (2017) assume access to a safe set of environment states, and a default safe policy, while in Fisac et al. (2018); Dean et al. (2019), knowledge of system dynamics is assumed and (Fisac et al., 2019) assume access to a distance metric on the state space. SAVED (Thananjeyan et al., 2020) learns a kernel density estimate over unsafe states, and assumes access to a set of user demonstrations and a user specified function that can be queried to determine whether a state is safe or not. In contrast to these approaches, our method does not assume any prior knowledge from the user, or domain knowledge of the problem setting, except a binary signal from the environment indicating when a catastrophic failure has occurred.

**Assuming a continuous safety cost function.** CPO (Achiam et al., 2017), and (Chow et al., 2019) assume a cost function can be queried from the environment at every time-step and the objective is to keep the cumulative costs within a certain limit. This assumption limits the generality of the method in scenarios where only minimal feedback, such as binary reward feedback is provided (additional details in section A.3).

(Stooke et al., 2020) devise a general modification to the Lagrangian by incorporating two additional terms in the optimization of the dual variable. SAMBA (Cowen-Rivers et al., 2020) has a learned GP dynamics model and a continuous constraint cost function that encodes safety. The objective is to minimize task cost function while maintaining the $\text{CVAR}_\alpha$ of cumulative costs below a threshold. In the work of Dalal et al. (2018); Paternain et al. (2019b;a); Grbic & Risi (2020), only the optimal policy is learned to be safe, and there are no safety constraint satisfactions during training. In contrast to these approaches, we assume only a binary signal from the environment indicating when a catastrophic failure has occurred. Instead of minimizing expected costs, our constraint formulation directly seeks to constrain the expected probability of failure.

**Safety through recoverability.** Prior works have attempted to devise resetting mechanisms to recover the policy to a base configuration from (near) a potentially unsafe state. LNT (Eysenbach et al., 2017) trains both a forward policy for solving a task, and a reset goal-conditioned policy that kicks in when the agent is in an *unsafe* state and learns an ensemble of critics, which is substantially more complex than our approach of a learned safety critic, which can give rise to a simple but provable safe exploration algorithm. Concurrently to us, SQRL (Srinivasan et al., 2020) developed an approach also using safety critics such that during the pre-training phase, the agent explores both safe and unsafe states in the environment for training the critic.

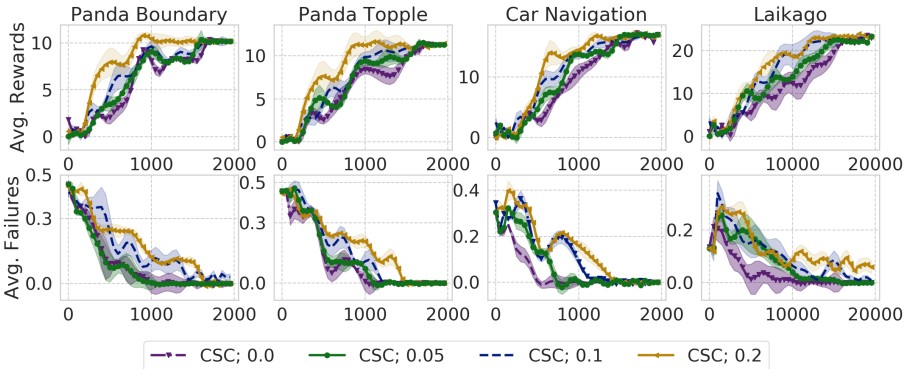

Figure 4: **Top row:** Average task rewards (higher is better). **Bottom row:** Average catastrophic failures (lower is better). **x-axis:** Number of episodes (each episode has 500 steps). Results on four of the five environments we consider for our experiments. For each environment we plot the average task reward, the average episodic failures, and the cumulative episodic failures. All the plots are for our method (CSC) with different safety thresholds $\chi$, specified in the legend. From the plots it is evident that our method can naturally trade-off safety for task performance depending on how strict the safety threshold is set to. Results are over four random seeds. Detailed results including plots of cumulative failures are in Fig. 5 of the Appendix.

In control theory, a number of prior works have focused on Hamilton-Jacobi-Isaacs (HJI) reachability analysis (Bansal et al., 2017) for providing safety constraint satisfactions and obtaining control inputs for dynamical systems (Herbert et al., 2019; Bajcsy et al., 2019; Leung et al., 2018). Our method does not require knowledge of the system dynamics or regularity conditions on the state-space, which are crucial for computing unsafe states using HJI reachability.

# 7 DISCUSSION, LIMITATIONS, AND CONCLUSION

We introduced a safe exploration algorithm to learn a conservative safety critic that estimates the probability of failure for each candidate state-action tuple, and uses this to constrain policy evaluation and policy improvement. We provably demonstrated that the probability of failures is bounded throughout training and provided convergence results showing how ensuring safety does not severely bottleneck task performance. We empirically validated our theoretical results and showed that we achieve high task performance while incurring low accidents during training.

While our theoretical results demonstrated that the probability of failures is bounded with a high probability, one limitation is that we still observe non-zero failures empirically even when the threshold $\chi$ is set to 0. This is primarily because of neural network function approximation error in the early stages of training the safety critic, which we cannot account for precisely in the theoretical results, and also due to the fact that we bound the *probability* of failures, which in practice means that the *number* of failures is also bounded, but non-zero. We also showed that if we set the constraint threshold in an appropriate time-varying manner, training with CSC incurs cumulative failures that scales at most sub-linearly with the number of transition samples in the environment.

Although our approach bounds the probability of failure and is general in the sense that it does not assume access any user-specified constraint function, in situations where the task is difficult to solve, for example due to stability concerns of the agent, our approach will fail without additional assumptions. In such situations, some interesting future work directions would be to develop a curriculum of tasks to start with simple tasks where safety is easier to achieve, and gradually move towards more difficult tasks, such that the learned knowledge from previous tasks is not forgotten.

## ACKNOWLEDGEMENT

We thank Vector Institute, Toronto and the Department of Computer Science, University of Toronto for compute support. We thank Glen Berseth and Kevin Xie for helpful initial discussions about the project, Alexandra Volokhova, Arthur Allshire, Mayank Mittal, Samarth Sinha, and Irene Zhang for feedback on the paper, and other members of the UofT CS Robotics Group for insightful discussions during internal presentations and reading group sessions. Finally, we are grateful to the anonymous ICLR 2021 reviewers for their feedback in helping improve the paper.

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

## A  APPENDIX

### A.1  PROOFS OF ALL THEOREMS AND LEMMAS

**Note.** During policy updates via Equation 5, the $D_{\text{KL}}$ constraint is satisfied with high probability if we follow Algorithm 1.

Following the steps in the Appendix A.2, we can write the **gradient ascent step for** $\phi$ as

$$\phi \leftarrow \phi_{old} + \beta \boldsymbol{F}^{-1} \nabla_{\phi_{old}} \tilde{J}(\phi_{old}) \quad \beta = \beta^j \sqrt{\frac{2\delta}{\nabla_{\phi_{old}} \tilde{J}(\phi_{old})^T \boldsymbol{F} \nabla_{\phi_{old}} \tilde{J}(\phi_{old})}} \tag{8}$$

$\boldsymbol{F}$ can be estimated with samples as

$$\boldsymbol{F} = \mathbb{E}_{s \sim \rho_{\phi_{old}}} \left[ \mathbb{E}_{a \sim \pi_{\phi_{old}}} \left[ \nabla_{\phi_{old}} \log \pi_{\phi_{old}} (\nabla_{\phi_{old}} \log \pi_{\phi_{old}})^T \right] \right] \tag{9}$$

Here $\beta^j$ is the backtracking coefficient and we perform backtracking line search with exponential decay. $\nabla_{\phi_{old}} \tilde{J}(\phi_{old})$ is calculated as,

$$\nabla_{\phi_{old}} \tilde{J}(\phi_{old}) = \mathbb{E}_{s \sim \rho_{\phi_{old}}, a \sim \pi_{\phi_{old}}} \left[ \nabla_{\phi_{old}} \log \pi_{\phi_{old}}(a|s) \tilde{A}_R^{\pi_{\phi_{old}}} \right] \tag{10}$$

After every update, we check if $\bar{D}_{\text{KL}}(\phi \| \phi_{old}) \leq \delta$, and if not we decay $\beta^j = \beta^j (1 - \beta^j)^j$, set $j \leftarrow j + 1$ and repeat for $L$ steps until $\bar{D}_{\text{KL}} \leq \delta$ is satisfied. If this is not satisfied after $L$ steps, we backtrack, and do not update $\phi$ i.e. set $\phi \leftarrow \phi_{old}$.

**Lemma 1.** *If we follow Algorithm 1, during policy updates via equation 5, the following is satisfied with high probability $\geq 1 - \omega$*

$$V_C^{\pi_{\phi_{old}}}(\mu) + \frac{1}{1-\gamma} \mathbb{E}_{s \sim \rho_{\phi_{old}}, a \sim \pi_\phi} [A_C(s,a)] \leq \chi + \zeta - \frac{\Delta}{1-\gamma}$$

*Here, $\zeta$ captures sampling error in the estimation of $V_C^{\pi_{\phi_{old}}}(\mu)$ and we have $\zeta \leq \frac{C\sqrt{\log(1/\omega)}}{|N|}$, where $C$ is a constant and $N$ is the number of samples used in the estimation of $V_C$.*

*Proof.* Based on line 6 of Algorithm 1, for every rollout $\{(s,a)\}$, the following holds:

$$\begin{aligned}
&Q_C(s,a) \leq (1-\gamma)(\chi - \hat{V}_C^{\pi_{\phi_{old}}}(\mu))) \quad \forall (s,a) \\
&\implies \hat{A}_C(s,a) \leq (1-\gamma)(\chi - \hat{V}_C^{\pi_{\phi_{old}}}(\mu))) \quad \forall (s,a) \\
&\implies \hat{V}_C^{\pi_{\phi_{old}}}(\mu) + \frac{1}{1-\gamma} \hat{A}_C(s,a) \leq \chi \quad \forall (s,a) \\
&\implies \hat{V}_C^{\pi_{\phi_{old}}}(\mu) + \frac{1}{1-\gamma} \mathbb{E}_{s \sim \rho_{\phi_{old}}, a \sim \pi_\phi} \left[ \hat{A}_C(s,a) \right] \leq \chi
\end{aligned} \tag{11}$$

We note that we can only compute a sample estimate $\hat{V}_C^{\pi_{\phi_{old}}}(\mu)$ instead of the true quantity $V_C$ which can introduce *sampling error* in practice. In order to ensure that $\hat{V}_C^{\pi_{\phi_{old}}}(\mu)$ is not much lesser than $V_C^{\pi_{\phi_{old}}}(\mu)$, we can obtain a bound on their difference. Note that if $\hat{V}_C^{\pi_{\phi_{old}}}(\mu) \geq V_C^{\pi_{\phi_{old}}}(\mu)$, the Lemma holds directly, so we only need to consider the less than case.

Let $\hat{V}_C^{\pi_{\phi_{old}}}(\mu) = V_C^{\pi_{\phi_{old}}}(\mu) - \zeta$. With high probability $\geq 1 - \omega$, we can ensure $\zeta \leq \frac{C'\sqrt{\log(1/\omega)}}{|N|}$, where $C'$ is a constant independent of $\omega$ (obtained from union bounds and concentration inequalities) and $N$ is the number of samples used in the estimation of $V_C$. In addition, our estimate of $\mathbb{E}_{s \sim \rho_{\phi_{old}}, a \sim \pi_\phi} \left[ \hat{A}_C(s,a) \right]$ is an overestimate of the true $\mathbb{E}_{s \sim \rho_{\phi_{old}}, a \sim \pi_\phi} [A_C(s,a)]$, and we denote their difference by $\Delta$.

So, with high probability $\geq 1 - \omega$, we have

$$\begin{aligned}
&\hat{V}_C^{\pi_{\phi_{old}}}(\mu) + \frac{1}{1-\gamma} \mathbb{E}_{s \sim \rho_{\phi_{old}}, a \sim \pi_\phi} \left[ \hat{A}_C(s,a) \right] \leq \chi \\
&\implies V_C^{\pi_{\phi_{old}}}(\mu) + \frac{1}{1-\gamma} \mathbb{E}_{s \sim \rho_{\phi_{old}}, a \sim \pi_\phi} [A_C(s,a)] \leq \chi + \zeta - \frac{\Delta}{1-\gamma}
\end{aligned} \tag{12}$$

$\square$

**Theorem 1.** *Consider policy updates that solve the constrained optimization problem defined in equation 5. With high probability $\geq 1 - \omega$, we have the following upper bound on expected probability of failure $V_C^{\pi_{\phi_{new}}}(\mu)$ for $\pi_{\phi_{new}}$ during every policy update iteration*

$$V_C^{\pi_{\phi_{new}}}(\mu) \leq \chi + \zeta - \frac{\Delta}{1 - \gamma} + \frac{\sqrt{2\delta}\gamma\epsilon_C}{(1 - \gamma)^2} \quad where \quad \zeta \leq \frac{C\sqrt{\log(1/\omega)}}{|N|} \quad (13)$$

Here, $\epsilon_C = \max_s |\mathbb{E}_{a \sim \pi_{\phi_{new}}} A_C(s,a)|$ and $\Delta$ is the overestimation in $\mathbb{E}_{s \sim \rho_{\phi_{old'}}, a \sim \pi_{\phi_{old}}}[A_C(s,a)]$ due to CQL.

*Proof.* $C(s)$ denotes the value of the constraint function from the environment in state $s$. This is analogous to the task reward function $R(s,a)$. In our case $C(s)$ is a binary indicator of whether a catastrophic failure has occurred, however the analysis we present holds even when $C(s)$ is a shaped continuous cost function.

$$C(s) = \begin{cases} 1, & \mathbb{1}\{failure\} = 1 \\ 0, & \text{otherwise} \end{cases}$$

Let $V_R^{\pi_\phi}(\mu)$ denotes the discounted task rewards obtained in expectation by executing policy $\pi_\phi$ for one episode, and let $V_C^{\pi_\phi}(\mu)$ denote the corresponding constraint values.

$$\max_{\pi_\phi} V_R^{\pi_\phi}(\mu) \quad s.t. \quad V_C^{\pi_\phi}(\mu) \leq \chi \quad (14)$$

From the TRPO (Schulman et al., 2015a) and CPO (Achiam et al., 2017) papers, following similar derivations, we obtain the following bounds

$$V_R^{\pi_\phi}(\mu) - V_R^{\pi_{\phi_{old}}}(\mu) \geq \frac{1}{1 - \gamma}\mathbb{E}_{s \sim \rho_{\phi_{old}}, a \sim \pi_\phi}\left[A_R^{\pi_{\phi_{old}}}(s,a) - \frac{2\gamma\epsilon_R}{1 - \gamma}D_{TV}(\pi_\phi||\pi_{\phi_{old}})[s]\right] \quad (15)$$

Here, $A_R^{\pi_\phi}$ is the advantage function corresponding to the task rewards and $\epsilon_R = \max_s |\mathbb{E}_{a \sim \pi_\phi} A_R^{\pi_\phi}(s,a)|$. $D_{TV}$ is the total variation distance. We also have,

$$V_C^{\pi_\phi}(\mu) - V_C^{\pi_{\phi_{old}}}(\mu) \leq \frac{1}{1 - \gamma}\mathbb{E}_{s \sim \rho_{\phi_{old}}, a \sim \pi_\phi}\left[A_C^{\pi_{\phi_{old}}}(s,a) + \frac{2\gamma\epsilon_C}{1 - \gamma}D_{TV}(\pi_\phi||\pi_{\phi_{old}})[s]\right] \quad (16)$$

Here, $A_C^{\pi_{\phi_{old}}}$ is the advantage function corresponding to the costs and $\epsilon_C = \max_s |\mathbb{E}_{a \sim \pi_\phi} A_C^{\pi_{\phi_{old}}}(s,a)|$. In our case, $A_C$ is defined in terms of the safety Q function $Q_C(s,a)$, and CQL can bound its expectation directly. To see this, note that, by definition $\mathbb{E}_{s \sim \rho_{\phi_{old}}, a \sim \pi_\phi}\left[A_C^{\pi_{\phi_{old}}}(s,a)\right] = \mathbb{E}_{s \sim \rho_{\phi_{old}}, a \sim \pi_\phi}[Q_\zeta(s,a)] - \mathbb{E}_{s \sim \rho_{\phi_{old}}, a \sim \pi_{\phi_{old}}}[Q_\zeta(s,a)]$. Here, the RHS is precisely the term in equation 2 of (Kumar et al., 2020) that is bounded by CQL. We get an overstimated advantage $\hat{A}_C(s,a)$ from training the safety critic $Q_C$ through updates in equation 2. . Let $\Delta$ denote the expected magnitude of over-estimate $\mathbb{E}_{s \sim \rho_{\phi_{old}}, a \sim \pi_\phi}\left[\hat{A}_C(s,a)\right] = \mathbb{E}_{s \sim \rho_{\phi_{old}}, a \sim \pi_\phi}[A_C(s,a)] + \Delta$, where $\Delta$ is positive. Note that replacing $A_C$, by its over-estimate $\hat{A}_C$, the inequality in 16 above still holds.

Using Pinsker's inequality, we can convert the bounds in terms of $D_{KL}$ instead of $D_{TV}$,

$$D_{TV}(p||q) \leq \sqrt{D_{KL}(p||q)/2} \quad (17)$$

By Jensen's inequality,

$$\mathbb{E}[\sqrt{D_{KL}(p||q)/2}] \leq \sqrt{\mathbb{E}[D_{KL}(p||q)]/2} \quad (18)$$

So, we can replace the $\mathbb{E}[D_{TV}(p||q)]$ terms in the bounds by $\sqrt{\mathbb{E}[D_{KL}(p||q)]}$. Then, inequation 16 becomes,

$$V_C^{\pi_\phi}(\mu) - V_C^{\pi_{\phi_{old}}}(\mu) \leq \frac{1}{1 - \gamma}\left[\mathbb{E}_{s \sim \rho_{\phi_{old}}, a \sim \pi_\phi}\left[A_C^{\pi_{\phi_{old}}}(s,a)\right] + \frac{2\gamma\epsilon_C}{1 - \gamma}\sqrt{\mathbb{E}_{s \sim \rho_{\phi_{old}}, a \sim \pi_\phi}[D_{KL}(\pi_\phi||\pi_{\phi_{old}})[s]]}\right]$$
$$(19)$$

Re-visiting our objective in equation 5,

$$\max_{\pi_\phi} \mathbb{E}_{s\sim\rho_{\phi_{old}},a\sim\pi_\phi}\left[A_R^{\pi_{\phi_{old}}}(s,a)\right]$$
$$s.t. \quad \mathbb{E}_{s\sim\rho_{\phi_{old}}}[D_{\mathrm{KL}}(\pi_{\phi_{old}}(\cdot|s)||\pi_\phi(\cdot|s))] \le \delta \tag{20}$$
$$s.t. \quad V_C^{\pi_\phi}(\mu) \le \chi$$

From inequation 19 we note that instead of of constraining $V_C^{\pi_\phi}(\mu)$ we can constrain an upper bound on this. Writing the constraint in terms of the current policy iterate $\pi_{\phi_{old}}$ using equation 19,

$$\pi_{\phi_{new}} = \max_{\pi_\phi} \mathbb{E}_{s\sim\rho_{\phi_{old}},a\sim\pi_\phi}\left[A_R^{\pi_{\phi_{old}}}(s,a)\right]$$
$$s.t. \quad \mathbb{E}_{s\sim\rho_{\phi_{old}}}[D_{\mathrm{KL}}(\pi_{\phi_{old}}(\cdot|s)||\pi_\phi(\cdot|s))] \le \delta$$
$$s.t. \quad V_C^{\pi_{\phi_{old}}}(\mu) + \frac{1}{1-\gamma}\mathbb{E}_{s\sim\rho_{\phi_{old}},a\sim\pi_\phi}\left[A_C^{\pi_{\phi_{old}}}(s,a)\right] + \beta\sqrt{\mathbb{E}_{s\sim\rho_{\phi_{old}}}[D_{\mathrm{KL}}(\pi_{\phi_{old}}(\cdot|s)||\pi_\phi(\cdot|s))]} \le \chi \tag{21}$$

As there is already a bound on $D_{\mathrm{KL}}(\pi_{\phi_{old}}(\cdot|s)||\pi_\phi(\cdot|s))]$, getting rid of the redundant term, we define the following optimization problem, which we actually optimize for

$$\pi_{\phi_{new}} = \max_{\pi_\phi} \mathbb{E}_{s\sim\rho_{\phi_{old}},a\sim\pi_\phi}\left[A_R^{\pi_{\phi_{old}}}(s,a)\right]$$
$$s.t. \quad \mathbb{E}_{s\sim\rho_{\phi_{old}}}[D_{\mathrm{KL}}(\pi_{\phi_{old}}(\cdot|s)||\pi_\phi(\cdot|s))] \le \delta \tag{22}$$
$$s.t. \quad V_C^{\pi_{\phi_{old}}}(\mu) + \frac{1}{1-\gamma}\mathbb{E}_{s\sim\rho_{\phi_{old}},a\sim\pi_\phi}\left[A_C^{\pi_{\phi_{old}}}(s,a)\right] \le \chi$$

**Upper bound on expected probability of failures.** If $\pi_{\phi_{new}}$ is updated using equation 5, then we have the following upper bound on $V_C^{\pi_{\phi_{new}}}(\mu)$

$$V_C^{\pi_{\phi_{new}}}(\mu) \le V_C^{\pi_{\phi_{old}}}(\mu) + \frac{1}{1-\gamma}\mathbb{E}_{s\sim\rho_{\phi_{old}},a\sim\pi_\phi}\left[A_C^{\pi_{\phi_{old}}}\right] + \frac{2\gamma\epsilon_C}{(1-\gamma)^2}\sqrt{\mathbb{E}_{s\sim\rho_{\phi_{old}},a\sim\pi_\phi}[\mathbf{D}_{KL}(\pi_\phi||\pi_{\phi_{old}})[s]]} \tag{23}$$

If we ensure $V_C^{\pi_{\phi_{old}}}(\mu) + \frac{1}{1-\gamma}\mathbb{E}_{s\sim\rho_{\phi_{old}},a\sim\pi_\phi}\left[A_C^{\pi_{\phi_{old}}}(s,a)\right] \le \chi$ holds by following Algorithm 1,we have the following upper bound on $V_C^{\pi_{\phi_{new}}}(\mu)$

$$V_C^{\pi_{\phi_{new}}}(\mu) \le \chi + \frac{\sqrt{2\delta}\gamma\epsilon_C}{(1-\gamma)^2} \tag{24}$$

Here, $\epsilon_C = \max_s |\mathbb{E}_{a\sim\pi_{\phi_{new}}}A_C^{\pi_{\phi_{old}}}(s,a)|$.

Now, instead of $A_C(s,a)$, we have an over-estimated advantage estimate $\hat{A}_C(s,a)$ obtained by training the safety critic $Q_C$ through CQL as in equation 2. Let $\Delta$ denote the expected magnitude of over-estimate $\mathbb{E}_{s\sim\rho_{\phi_{old}},a\sim\pi_\phi}\left[\hat{A}_C(s,a)\right] = \mathbb{E}_{s\sim\rho_{\phi_{old}},a\sim\pi_\phi}[A_C(s,a)] + \Delta$, where $\Delta$ is positive.

From Lemma 1, we are able to ensure the following with high probability $\ge 1 - \omega$

$$V_C^{\pi_{\phi_{old}}}(\mu) + \frac{1}{1-\gamma}\mathbb{E}_{s\sim\rho_{\phi_{old}},a\sim\pi_\phi}[A_C(s,a)] \le \chi + \zeta - \frac{\Delta}{1-\gamma}$$

By combining this with the upper bound on $V_C^{\pi_{\phi_{new}}}(\mu)$ from inequality 23, we obtain with probability $\ge 1 - \omega$

$$V_C^{\pi_{\phi_{new}}}(\mu) \le \chi + \zeta - \frac{\Delta}{1-\gamma} + \frac{\sqrt{2\delta}\gamma\epsilon_C}{(1-\gamma)^2} \quad \text{where} \quad \zeta \le \frac{C'\sqrt{\log(1/\omega)}}{|N|} \tag{25}$$

$\square$

Since $\epsilon_C$ depends on the optimized policy $\pi_{\phi_{new}}$, it can't be calculated exactly prior to the update. As we cap $Q_C(s,a)$ to be $\le 1$, therefore, the best bound we can construct for $\epsilon_C$ is the trivial bound $\epsilon_C \le 2$. Now, in order to have $V_C^{\pi_{\phi_{new}}}(\mu) < \chi$, we require $\Delta > \frac{2\sqrt{2\delta}\gamma}{1-\gamma} + (1-\gamma)\zeta$. To guarantee this, replacing $\Delta$ by the exact overestimation term from CQL, we have the following condition on

$\alpha$:

$$\alpha > \frac{\mathcal{G}_{c,T}}{1-\gamma} \cdot \max_{s \sim \rho_{\phi_{old'}}} \left( \frac{1}{|\sqrt{\mathcal{D}_{\phi_{old'}}}|} + \frac{2\sqrt{2\delta}\gamma + (1-\gamma)^2 \zeta}{\mathcal{G}_{c,T}} \right) \left[ \mathbb{E}_{a \sim \pi_{\phi_{old}}} \left( \frac{\pi_{\phi_{old}}}{\pi_{\phi_{old'}}} - 1 \right) \right]^{-1} \quad (26)$$

Here, $\mathcal{G}_{c,T}$ is a constant depending on the concentration properties of the safety constraint function $C(s,a)$ and the state transition operator $T(s'|s,a)$ (Kumar et al., 2020). $\phi_{old'}$ denotes the parameters of the policy $\pi$ in the iteration before $\phi_{old}$. Now, with probability $\geq 1 - \omega$, we have $\zeta \leq \frac{C' \sqrt{\log(1/\omega)}}{|N|}$. So, if $\alpha$ is chosen as follows

$$\alpha > \frac{\mathcal{G}_{c,T}}{1-\gamma} \cdot \max_{s \sim \rho_{\phi_{old'}}} \left( \frac{1}{|\sqrt{\mathcal{D}_{\phi_{old'}}}|} + \frac{2\sqrt{2\delta}\gamma + (1-\gamma)^2 \frac{C' \sqrt{\log(1/\omega)}}{|N|}}{\mathcal{G}_{c,T}} \right) \left[ \mathbb{E}_{a \sim \pi_{\phi_{old}}} \left( \frac{\pi_{\phi_{old}}}{\pi_{\phi_{old'}}} - 1 \right) \right]^{-1}$$

$$(27)$$

Then with probability $\geq 1 - \omega$, we will have,

$$V_C^{\pi_{\phi_{new}}}(\mu) \leq \chi \quad (28)$$

In the next theorem, we show that the convergence rate to the optimal solution is not severely affected due to the safety constraint satisfaction guarantee, and gets modified by addition of an extra bounded term.

**Theorem 2.** *If we run the policy gradient updates through equation 5, for policy $\pi_\phi$, with $\mu$ as the starting state distribution, with $\phi^{(0)} = 0$, and learning rate $\eta > 0$, then for all policy update iterations $T > 0$ we have, with probability $\geq 1 - \omega$,*

$$V_R^*(\mu) - V_R^{(T)}(\mu) \leq \frac{\log |\mathcal{A}|}{\eta T} + \frac{1}{(1-\gamma)^2 T} + \left( (1-\chi) + \left( 1 - \frac{2\Delta}{(1-\gamma)} \right) + 2\zeta \right) \frac{\sum_{t=0}^{T-1} \lambda^{(t)}}{\eta T}$$

Since the value of the dual variables $\lambda$ strictly decreases during gradient descent updates (Algorithm 1), $\sum_{t=0}^{T-1} \lambda^{(t)}$ is upper-bounded. In addition, if we choose $\alpha$ as mentioned in the discussion of Theorem 1, we have $\Delta > \frac{2\sqrt{2\delta}\gamma}{1-\gamma} + \zeta$. Hence, with probability $\geq 1 - \omega$, we can ensure that

$$V_R^*(\mu) - V_R^{(T)}(\mu) \leq \frac{\log |\mathcal{A}|}{\eta T} + \frac{1}{(1-\gamma)^2 T} + K \frac{\sum_{t=0}^{T-1} \lambda^{(t)}}{\eta T} \quad \text{where} \quad K \leq (1-\chi) + \frac{4\sqrt{2\delta}\gamma}{(1-\gamma)^2}$$

*Proof.* Let superscript $(t)$ denote the $t^{\text{th}}$ policy update iteration. We follow the derivation in Lemma 5.2 of (Agarwal et al., 2019) but replace $A(s,a)$ with our modified advantage estimator $\hat{A}^{(t)}(s,a) = A_R^{(t)}(s,a) - \lambda^{(t)} A_C(s,a)$. The quantity $\log Z_t(s)$ is defined in terms of $A_R^{(t)}$ as

$$\log Z_t(s) = \log \sum_a \pi^{(t)}(a|s) \exp \left( \eta A^{(t)}/(1-\gamma) \right)$$

$$\geq \sum_a \pi^{(t)}(a|s) \log \exp \eta A^{(t)}(s,a)/(1-\gamma)) \quad (29)$$

$$= \frac{\eta}{1-\gamma} \sum_a \pi^{(t)}(a|s) A^{(t)}(s,a)$$

$$= 0$$

We define an equivalent alternate quantity based on $\hat{A}^{(t)}$

$$\log \hat{Z}_t(s) = \log \sum_a \pi^{(t)}(a|s) \exp \left( \eta \hat{A}^{(t)}(s,a)/(1-\gamma) \right)$$

$$= \log \sum_a \pi^{(t)}(a|s) \exp \left( \eta (A_R^{(t)}(s,a) - \lambda^{(t)} A_C(s,a))/(1-\gamma) \right)$$

$$\geq \sum_a \pi^{(t)}(a|s) \log \exp \left( \eta A_R^{(t)}(s,a)/(1-\gamma) \right) - \lambda^{(t)} \sum_a \pi^{(t)}(a|s) \log \exp \left( \eta A_C^{(t)}(s,a)/(1-\gamma) \right)$$

$$= 0 - \frac{\lambda^{(t)} \eta}{1-\gamma} \sum_a \pi^{(t)}(a|s) A_C^{(t)}(s,a)$$

$$(30)$$

For simplicity, consider softmax policy parameterization (equivalent results hold under the function approximation regime as shown in (Agarwal et al., 2019)), where we define the policy updates with the modified advantage function $\hat{A}^{(t)}$ to take the form:

$$\phi^{(t+1)} = \phi^{(t)} + \frac{\eta}{1-\gamma}\hat{A}^{(t)} \quad \text{and} \quad \pi^{(t+1)}(a|s) = \pi^{(t)}(a|s)\frac{\exp(\eta\hat{A}^{(t)}(s,a)/(1-\gamma))}{\hat{Z}_t(s)},$$

Here, $\hat{Z}_t(s) = \sum_{a \in \mathcal{A}} \pi^{(t)}(a|s)\exp(\eta\hat{A}^{(t)}(s,a)/(1-\gamma))$. Note that our actual policy updates (with backtracking line search) are almost equivalent to this when $\eta$ is *small*. For the sake of notational convenience, we will denote $\log \hat{Z}_t(s) + \frac{\lambda^{(t)}\eta}{1-\gamma}\sum_a \pi^{(t)}(a|s)A_C^{(t)}(s,a)$ as $G_t(s)$. We have $G_t(s) \geq 0$ from equation 30.

We consider the performance improvement lemma (Kakade & Langford, 2002) with respect to the task advantage function $A_R^{(t)}(s,a)$ and express it in terms of the modified advantage function $\hat{A}^{(t)}(s,a) = A_R^{(t)}(s,a) - \lambda^{(t)}A_C(s,a)$. Let $\mu$ be the starting state distribution of the MDP, and $d^{(t)}$ denote the stationary distribution of states induced by policy $\pi$ in the $t^{\text{th}}$ iteration.

$$
\begin{aligned}
V_R^{(t+1)}(\mu) - V_R^{(t)}(\mu) &= \frac{1}{1-\gamma}\mathbb{E}_{s \sim d^{(t+1)}}\sum_a \pi^{(t+1)}(a|s)A_R^{(t)}(s,a) \\
&= \frac{1}{1-\gamma}\mathbb{E}_{s \sim d^{(t+1)}}\sum_a \pi^{(t+1)}(a|s)(\hat{A}^{(t)}(s,a) + \lambda^{(t)}A_C^{(t)}(s,a)) \\
&= \frac{1}{\eta}\mathbb{E}_{s \sim d^{(t+1)}}\sum_a \pi^{(t+1)}(a|s)\log\frac{\pi^{(t+1)}(a|s)\hat{Z}_t(s)}{\pi^{(t)}(a|s)} \\
&\quad + \frac{1}{1-\gamma}\mathbb{E}_{s \sim d^{(t+1)}}\sum_a \pi^{(t+1)}(a|s)(\lambda^{(t)}A_C^{(t)}(s,a)) \\
&= \frac{1}{\eta}\mathbb{E}_{s \sim d^{(t+1)}}D_{\text{KL}}(\pi_s^{(t+1)}||\pi_s^{(t)}) + \frac{1}{\eta}\mathbb{E}_{s \sim d^{(t+1)}}\log\hat{Z}_t(s) \\
&\quad + \frac{1}{1-\gamma}\mathbb{E}_{s \sim d^{(t+1)}}\sum_a \pi^{(t+1)}(a|s)(\lambda^{(t)}A_C^{(t)}(s,a)) \\
&\geq \frac{1}{\eta}\mathbb{E}_{s \sim d^{(t+1)}}\log\hat{Z}_t(s) + \frac{\lambda^{(t)}}{1-\gamma}\mathbb{E}_{s \sim d^{(t+1)}}\sum_a \pi^{(t)}(a|s)A_C^{(t)}(s,a) \\
&\geq \frac{1}{\eta}\mathbb{E}_{s \sim d^{(t+1)}}G_t(s) \\
&\geq \frac{1-\gamma}{\eta}\mathbb{E}_{s \sim \mu}G_t(s)
\end{aligned}
\tag{31}
$$

We note that $G_t(s) \geq 0$ from equation 30. We now prove a result upper bounding the difference between the optimal task value for any state distribution $\rho$ and the task value at the $t^{\text{th}}$ iteration for the same state distribution.

**Sub-optimality gap.** The difference between the optimal value function and the current value function estimate is upper bounded.

$$V_R^{\pi^\star}(\rho) - V_R^{(t)}(\rho) = \frac{1}{1-\gamma} \mathbb{E}_{s \sim d^\star} \sum_a \pi^\star(a|s)(\hat{A}^{(t)}(s,a) + \lambda^{(t)} A_C^{(t)}(s,a))$$

$$= \frac{1}{\eta} \mathbb{E}_{s \sim d^\star} \sum_a \pi^\star(a|s) \log \frac{\pi^{(t+1)}(a|s)\hat{Z}_t(s)}{\pi^{(t)}(a|s)} + \frac{1}{1-\gamma} \mathbb{E}_{s \sim d^\star} \sum_a \pi^\star(a|s)\lambda^{(t)} A_C^{(t)}(s,a)$$

$$= \frac{1}{\eta} \mathbb{E}_{s \sim d^\star} \left( D_{\mathrm{KL}}(\pi_s^\star||\pi_s^{(t)}) - D_{\mathrm{KL}}(\pi_s^\star||\pi_s^{(t+1)}) + \sum_a \pi^*(a|s) \log \hat{Z}_t(s) \right)$$

$$+ \frac{1}{1-\gamma} \mathbb{E}_{s \sim d^\star} \sum_a \pi^\star(a|s)\lambda^{(t)} A_C^{(t)}(s,a)$$

$$= \frac{1}{\eta} \mathbb{E}_{s \sim d^\star} \left( D_{\mathrm{KL}}(\pi_s^\star||\pi_s^{(t)}) - D_{\mathrm{KL}}(\pi_s^\star||\pi_s^{(t+1)}) + \log \hat{Z}_t(s) \right) + \frac{1}{1-\gamma} \mathbb{E}_{s \sim d^\star} \sum_a \pi^\star(a|s)\lambda^{(t)} A_C^{(t)}(s,a)$$

$$= \frac{1}{\eta} \mathbb{E}_{s \sim d^\star} \left( D_{\mathrm{KL}}(\pi_s^\star||\pi_s^{(t)}) - D_{\mathrm{KL}}(\pi_s^\star||\pi_s^{(t+1)}) \right) + \frac{1}{\eta} \mathbb{E}_{s \sim d^\star} \left( \log \hat{Z}_t(s) + \frac{\lambda^{(t)}}{1-\gamma} \sum_a \pi^\star(a|s) A_C^{(t)}(s,a) \right)$$

$$= \frac{1}{\eta} \mathbb{E}_{s \sim d^\star} \left( D_{\mathrm{KL}}(\pi_s^\star||\pi_s^{(t)}) - D_{\mathrm{KL}}(\pi_s^\star||\pi_s^{(t+1)}) \right)$$

$$+ \frac{1}{\eta} \mathbb{E}_{s \sim d^\star} \left( G_t(s) + \frac{\lambda^{(t)}}{1-\gamma} \sum_a \pi^\star(a|s) A_C^{(t)}(s,a) - \frac{\lambda^{(t)}}{1-\gamma} \sum_a \pi^{(t)}(a|s) A_C^{(t)}(s,a) \right)$$
$$\tag{32}$$

Using equation 31 with $d^\star$ as the starting state distribution $\mu$, we have:
$$\frac{1}{\eta} \mathbb{E}_{s \sim d^\star} \log G_t(s) \le \frac{1}{1-\gamma} \left( V^{(t+1)}(d^\star) - V^{(t)}(d^\star) \right)$$
which gives us a bound on $\mathbb{E}_{s \sim d^\star} \log G_t(s)$.

Using the above equation and that $V^{(t+1)}(\rho) \ge V^{(t)}(\rho)$ (as $V^{(t+1)}(s) \ge V^{(t)}(s)$ for all states $s$), we have:

$$V_R^{\pi^\star}(\rho) - V_R^{(T-1)}(\rho) \le \frac{1}{T} \sum_{t=0}^{T-1} (V_R^{\pi^\star}(\rho) - V_R^{(t)}(\rho))$$

$$\le \frac{1}{\eta T} \sum_{t=0}^{T-1} \mathbb{E}_{s \sim d^\star}(D_{\mathrm{KL}}(\pi_s^\star||\pi_s^{(t)}) - D_{\mathrm{KL}}(\pi_s^\star||\pi_s^{(t+1)})) + \frac{1}{\eta T} \sum_{t=0}^{T-1} \mathbb{E}_{s \sim d^\star} \log G_t(s)$$

$$+ \frac{1}{\eta T} \sum_{t=0}^{T-1} \mathbb{E}_{s \sim d^\star} \left( \frac{\lambda^{(t)}}{1-\gamma} \sum_a \pi^\star(a|s) A_C^{(t)}(s,a) - \frac{\lambda^{(t)}}{1-\gamma} \sum_a \pi^{(t)}(a|s) A_C^{(t)}(s,a) \right)$$

$$\le \frac{\mathbb{E}_{s \sim d^\star} D_{\mathrm{KL}}(\pi_s^\star||\pi^{(0)})}{\eta T} + \frac{1}{(1-\gamma)T} \sum_{t=0}^{T-1} \left( V_R^{(t+1)}(d^\star) - V_R^{(t)}(d^\star) \right)$$

$$+ \frac{1}{\eta T} \sum_{t=0}^{T-1} \lambda^{(t)} \left( \frac{1}{1-\gamma} \mathbb{E}_{s \sim d^\star} \sum_a \pi^\star(a|s) A_C^{(t)}(s,a) - \frac{1}{1-\gamma} \mathbb{E}_{s \sim d^\star} \sum_a \pi^{(t)}(a|s) A_C^{(t)}(s,a) \right)$$

$$\le \frac{\mathbb{E}_{s \sim d^\star} D_{\mathrm{KL}}(\pi_s^\star||\pi^{(0)})}{\eta T} + \frac{V_R^{(T)}(d^\star) - V_R^{(0)}(d^\star)}{(1-\gamma)T} + 2((1-\gamma)(\chi+\zeta) - \Delta) \frac{\sum_{t=0}^{T-1} \lambda^{(t)}}{(1-\gamma)\eta T}$$

$$\le \frac{\log|\mathcal{A}|}{\eta T} + \frac{1}{(1-\gamma)^2 T} + 2((1-\gamma)(\chi+\zeta) - \Delta) \frac{\sum_{t=0}^{T-1} \lambda^{(t)}}{(1-\gamma)\eta T}.$$

Here, $\Delta$ denotes the CQL overestimation penalty, and we have used the fact that each term of $\left( \frac{1}{1-\gamma} \sum_a \pi^\star(a|s) A_C^{(t)}(s,a) - \frac{1}{1-\gamma} \sum_a \pi^{(t)}(a|s) A_C^{(t)}(s,a) \right)$ is upper bounded by $(\chi + \zeta - \frac{\Delta}{(1-\gamma)})$ from Lemma 1, so the difference is upper-bounded by $2(\chi + \zeta - \frac{\Delta}{(1-\gamma)})$.

By choosing $\alpha$ as in equation 26, we have $\Delta > \frac{2\sqrt{2\delta}\gamma}{1-\gamma} + (1-\gamma)\zeta$. So, $-\Delta < -\frac{2\sqrt{2\delta}\gamma}{1-\gamma} - (1-\gamma)\zeta$. Hence, we obtain the relation

We also observe that $2(\chi - \frac{\Delta}{(1-\gamma)}) + 2\zeta = \chi + \chi - 2\frac{\Delta}{(1-\gamma)} + 2\zeta \leq 2 - \chi - 2\frac{\Delta}{(1-\gamma)} = (1-\chi) + 2\zeta + (1 - 2\frac{\Delta}{(1-\gamma)}) + 2\zeta$

So, we have the following result for convergence rate

$$V_R^*(\mu) - V_R^{(T)}(\mu) \leq \frac{\log|\mathcal{A}|}{\eta T} + \frac{1}{(1-\gamma)^2 T} + ((1-\chi) + (1 - \frac{2\Delta}{(1-\gamma)}) + 2\zeta)\frac{\sum_{t=0}^{T-1}\lambda^{(t)}}{\eta T}$$

Again, with probability $\geq 1-\omega$, we can ensure $\zeta \leq \frac{C'\sqrt{\log(1/\omega)}}{|N|}$. Overall, choosing the value of $\alpha$ from equation 27, we have $\Delta > \frac{2\sqrt{2\delta}\gamma}{1-\gamma} + (1-\gamma)\zeta$. So, $-\Delta < -\frac{2\sqrt{2\delta}\gamma}{1-\gamma} - (1-\gamma)\zeta$. Hence, with probability $\geq 1-\omega$, we can ensure that

$$V_R^*(\mu) - V_R^{(T)}(\mu) \leq \frac{\log|\mathcal{A}|}{\eta T} + \frac{1}{(1-\gamma)^2 T} + K\frac{\sum_{t=0}^{T-1}\lambda^{(t)}}{\eta T}$$

where,

$$K \leq (1-\chi) + \frac{4\sqrt{2\delta}\gamma}{(1-\gamma)^2}$$

$\square$

So far we have demonstrated that the resulting policy iterates from our algorithm all satisfy the desired safety constraint of the CMDP which allows for a maximum safety violation of $\chi$ for every intermediate policy. While this guarantee ensures that the probability of failures is bounded, it does not elaborate on the total failures incurred by the algorithm. In our next result, we show that the cumulative failures until a certain number of iterations $T$ of the algorithm grows sublinearly when executing Algorithm 1, provided the safety threshold $\chi$ is set in the right way.

**Theorem 3.** *Let $\chi$ in Algorithm 1 be time-dependent such that $\chi_t = \mathcal{O}(1/\sqrt{t})$. Then, the total number of cumulative safety violations until when $T$ transition samples have been collected by Algorithm 1, $Reg_C(T)$, scales sub-linearly with $T$, i.e., $Reg_C(T) = \mathcal{O}(\sqrt{|\mathcal{S}||\mathcal{A}|T})$.*

*Proof.* From Theorem 1, with probability $\geq 1-\omega$

$$V_C^{\pi_{\phi new}}(\mu) \leq \chi + \zeta - \frac{\Delta}{1-\gamma} + \frac{\sqrt{2\delta}\gamma\epsilon_C}{(1-\gamma)^2} \qquad \text{where} \qquad \zeta \leq \frac{C'\sqrt{\log(1/\omega)}}{|N|} \qquad (33)$$

Instead of using $\phi_{new}$, for ease of notation in this analysis, let us write this in terms of subscript $t$ such that $\pi_t$ denotes the policy at the $t^{\text{th}}$ iteration. We can write the bound on $\zeta$ more explicitly as, $\zeta_t \leq \mathbb{E}_{s\sim d^{\pi_t}, a\sim\pi_t}\left[\frac{C'\sqrt{\log(1/\omega)}}{\sqrt{|N_t(s,a)|}}\right]$ Here, $N_t(s,a)$ denotes the total number of times $(s,a)$ is seen, and can be written as $N_t(s,a) = \sum_{j=1}^t n_j(s,a)$, where $n_j(s,a)$ denotes total number of times $(s,a)$ is seen in episode $j$. $T = \sum_{s,a} N_k(s,a)$ denotes the total number of samples collected so far. Let $k$ be the number of episodes elapsed when this happens. The safety threshold $\chi$ can be varied at every iteration, such that it decreases as $\chi_t \propto \frac{1}{\sqrt{t}}$. So, we have with probability $\geq 1-\omega$

$$V_C^{\pi_t}(\mu) \leq \chi_t + \zeta_t - \frac{\Delta}{1-\gamma} + \frac{\sqrt{2\delta}\gamma\epsilon_C}{(1-\gamma)^2} \quad \text{where} \quad \zeta_t \leq \mathbb{E}_{s\sim d^{\pi_t}, a\sim\pi_t}\left[\frac{C'\sqrt{\log(1/\omega)}}{\sqrt{|N_t(s,a)|}}\right] \quad \text{and} \quad \chi_t = \frac{C''}{\sqrt{t}}$$
$$(34)$$

Now, we note that $V_C^{\pi_t}$ denotes the expected probability of episodic failures by executing policy $\pi_t$ (as described in section 2 of the main paper). Let us consider that a policy is rolled out for one episode after every training iteration to collect data (exploration).

Hence the total cumulative safety failures when $T$ transition samples have been collected by Algorithm 1 is:

$$
\begin{aligned}
\mathrm{Reg}_C(T) = \sum_{t=1}^{k} V_C^{\pi_t}(\mu) \ \times \ 1 \\
\leq \sum_{t=1}^{k}(\chi_t + \zeta_t) - \sum_{t=1}^{k}\left(\frac{\Delta}{1-\gamma} - \frac{\sqrt{2\delta}\gamma\epsilon_C}{(1-\gamma)^2}\right) \\
\leq \sum_{t=1}^{k}\frac{C''}{\sqrt{t}} + \sum_{t=1}^{k}\sum_{s,a} d^{\pi_t}(s)\pi_t(a|s)\frac{C'\sqrt{\log(1/\omega)}}{\sqrt{|N_t(s,a)|}} \\
\leq \sum_{t=1}^{k}\frac{C''}{\sqrt{t}} + C'\sqrt{\log(1/\omega)}\sum_{s,a}\sum_{t=1}^{k}\frac{n_t(s,a)}{\sqrt{|N_t(s,a)|}} \\
\leq C''\sqrt{k} + C'\sqrt{\log(1/\omega)}\sum_{s,a}\sqrt{N_k(s,a)} \\
\leq C''\sqrt{T} + C'\sqrt{\log(1/\omega)}\sqrt{|\mathcal{S}||\mathcal{A}|}\sqrt{\sum_{s,a}N_k(s,a)} \\
\leq C'''\sqrt{|\mathcal{S}||\mathcal{A}|T} \\
= \mathcal{O}\sqrt{|\mathcal{S}||\mathcal{A}|T}
\end{aligned}
\tag{35}
$$

$\square$

*Here, we used the concavity of square root function to obtain* $\sum_{s,a}\sqrt{N_k(s,a)} \leq \sqrt{|\mathcal{S}||\mathcal{A}|}\sqrt{\sum_{s,a}N_k(s,a)}$ *and the definition* $T = \sum_{s,a}N_k(s,a)$

### A.2 Derivation of the policy update equations

Let $a \in \mathcal{A}$ denote an action, $s \in \mathcal{S}$ denote a state, $\pi_\phi(a|s)$ denote a parameterized policy, $r(s,a)$ denote a reward function for the task being solved, and $\tau$ denote a trajectory of actions by following policy $\pi_\phi$ at each state. To solve the following constrained optimization problem:

$$
\max_{\pi_\phi}\mathbb{E}_{\tau \sim \pi_\phi}[\sum_\tau r(\cdot)] \quad s.t. \quad \mathbb{E}_{\tau \sim \pi_\phi}[\sum_\tau \mathbb{1}\{failure\}] = 0
\tag{36}
$$

Here, $\tau$ is the trajectory corresponding to an episode. The objective is to maximize the cumulative returns while satisfying the constraint. The constraint says that the agent must never fail during every episode. $\mathbb{1}\{failure\} = 1$ if there is a failure and $\mathbb{1}\{failure\} = 0$ if the agent does not fail. The only way expectation can be $0$ for this quantity is if every element is $0$, so the constraint essentially is to never fail in any episode. Let's rewrite the objective, more generally as

$$
\max_{\pi_\phi} V_R^{\pi_\phi}(\mu) \quad s.t. \quad V_C^{\pi_\phi}(\mu) = 0
\tag{37}
$$

We can relax the constraint slightly, by introducing a tolerance parameter $\chi \approx 0$. The objective below tolerates atmost $\chi$ failures in expectation. Since the agent can fail only once in an episode, $V_C^{\pi_\phi}(\mu)$ can also be interpreted as the *probability of failure*, and the constraint $V_C^{\pi_\phi}(\mu) \leq \chi$ says that the probability of failure in expectation must be bounded by $\chi$. So, our objective has a very intuitive and practical interpretation.

$$
\max_{\pi_\phi} V_R^{\pi_\phi}(\mu) \quad s.t. \quad V_C^{\pi_\phi}(\mu) \leq \chi
\tag{38}
$$

We learn one state value function, $V_R$ (corresponding to the task reward), parameterized by $\theta$ and one state-action value function $Q_C$ (corresponding to the sparse failure indicator), parameterized by $\zeta$. We have a task reward function $r(s,a)$ from the environment which is used to learn $V_R$. For learning $Q_C$, we get a signal from the environment indicating whether the agent is dead (1) or alive (0) i.e. $\mathbb{1}\{failure\}$.

The safety critic $Q_C$ is used to get an estimate of how safe a particular state is, by providing an estimate of *probability of failure*, that will be used to guide exploration. We desire the estimates to be conservative, in the sense that the probability of failure should be an *over-estimate* of the actual probability so that the agent can err in the side of caution while exploring. To train such a critic $Q_C$, we incorporate theoretical insights from CQL, and estimate $Q_C$ through updates similar to those obtained by flipping the sign of $\alpha$ in equation 2 of the CQL paper (Kumar et al., 2020). The motivation for this is to get an *upper bound* on $Q_C$ instead of a lower bound, as guaranteed by CQL.

We also note that the CQL penalty term (the first two terms of equation 2 of the CQL paper) can be expressed as an estimate for the advantage function of the policy $\mathbb{E}_{s\sim d^{\pi_{\phi_{old}}}, a\sim\pi_\phi(a|s)}[A(s,a)]$, where, $A(s,a)$ is the advantage function.

$$\mathbb{E}_{s\sim d^{\pi_{\phi_{old}}}, a\sim\pi_\phi(a|s)}[Q(s,a)] - \mathbb{E}_{s\sim d^{\pi_{\phi_{old}}}, a\sim\pi_{\phi_{old}}(a|s)}[Q(s,a)]$$
$$= \mathbb{E}_{s\sim d^{\pi_{\phi_{old}}}, a\sim\pi_\phi(a|s)}[Q(s,a) - \mathbb{E}_{a\sim\pi_{\phi_{old}}(a|s)}Q(s,a)] \tag{39}$$
$$= \mathbb{E}_{s\sim d^{\pi_{\phi_{old}}}, a\sim\pi_\phi(a|s)}[Q(s,a) - V(s)] = \mathbb{E}_{s\sim d^{\pi_{\phi_{old}}}, a\sim\pi_\phi(a|s)}[A(s,a)]$$

Hence, CQL can help provide an upper bound on the advantage function directly. Although the CQL class of algorithms have been proposed for batch RL, the basic bounds on the value function hold even for online training.

We denote the objective inside $\arg\min$ as $CQL(\zeta)$, where $\zeta$ parameterizes $Q_C$, and $k$ denotes the $k^{\text{th}}$ update iteration.

$$\hat{Q}_C^{k+1} \leftarrow \underset{Q_C}{\arg\min}\, \alpha\left(-\mathbb{E}_{s\sim\mathcal{D}_{env}, a\sim\pi_\phi(a|s)}[Q_C(s,a)] + \mathbb{E}_{(s,a)\sim\mathcal{D}_{env}}[Q_C(s,a)]\right)$$
$$+ \frac{1}{2}\mathbb{E}_{(s,a,s',c)\sim\mathcal{D}_{env}}\left[\left(Q_C(s,a) - \hat{\mathcal{B}}^{\pi_\phi}\hat{Q}_C^k(s,a)\right)^2\right] \tag{40}$$

For states sampled from the replay buffer $\mathcal{D}_{env}$, the first term seeks to maximize the expectation of $Q_C$ over actions sampled from the current policy, while the second term seeks to minimize the expectation of $Q_C$ over actions sampled from the replay buffer. $\mathcal{D}_{env}$ can include off-policy data, and also offline-data (if available). Let the over-estimated advantage, corresponding to the over-estimated critic $Q_C$, so obtained from CQL, be denoted as $\hat{A}_C(s,a)$, where the *true* advantage is $A_C(s,a)$.

Now, let $\rho_\phi(s)$ denote the stationary distribution of states induced by policy $\pi_\phi$. For policy optimization, we have to solve a constrained optimization problem as described below:

$$\max_{\pi_\phi} \mathbb{E}_{s\sim\rho_{\phi_{old}}, a\sim\pi_\phi}\left[A_R^{\pi_{\phi_{old}}}(s,a)\right]$$
$$s.t.\quad \mathbb{E}_{s\sim\rho_{\phi_{old}}}[D_{\text{KL}}(\pi_{\phi_{old}}(\cdot|s)||\pi_\phi(\cdot|s))] \leq \delta \tag{41}$$
$$s.t.\quad V_C^{\pi_\phi}(\mu) \leq \chi$$

This, as per equation 22 can be rewritten as

$$\pi_{\phi_{new}} = \max_{\pi_\phi} \mathbb{E}_{s\sim\rho_{\phi_{old}}, a\sim\pi_\phi}\left[A_R^{\pi_{\phi_{old}}}(s,a)\right]$$
$$s.t.\quad \mathbb{E}_{s\sim\rho_{\phi_{old}}}[D_{\text{KL}}(\pi_{\phi_{old}}(\cdot|s)||\pi_\phi(\cdot|s))] \leq \delta \tag{42}$$
$$s.t.\quad V_C^{\pi_{\phi_{old}}}(\mu) + \frac{1}{1-\gamma}\mathbb{E}_{s\sim\rho_{\phi_{old}}, a\sim\pi_\phi}\left[A_C^{\pi_{\phi_{old}}}(s,a)\right] \leq \chi$$

Since we are learning an over-estimate of $A_C$ through the updates in equation 2, we replace $A_C$ by the learned $\hat{A}_C$ in the constraint above. There are multiple ways to solve this constrained optimization problem, through duality. If we consider the Lagrangian dual of this, then we have the following optimization problem, which we can solve *approximately* by alternating gradient descent. For now, we keep the KL constraint as is, and later use its second order Taylor expansion in terms of the Fisher Information Matrix.

$$\max_{\pi_\phi} \min_{\lambda \geq 0} \mathbb{E}_{s \sim \rho_{\phi_{old}}, a \sim \pi_\phi} \left[ A_R^{\pi_{\phi_{old}}}(s,a) \right] - \lambda \left( V_C^{\pi_{\phi_{old}}}(\mu) + \frac{1}{1-\gamma} \mathbb{E}_{s \sim \rho_{\phi_{old}}, a \sim \pi_\phi} \left[ \hat{A}_C(s,a) \right] - \chi \right)$$
$$s.t. \quad \mathbb{E}_{s \sim \rho_{\phi_{old}}} [D_{\text{KL}}(\pi_{\phi_{old}}(\cdot|s) || \pi_\phi(\cdot|s))] \leq \delta \tag{43}$$

We replace $V_C^{\pi_{\phi_{old}}}(\mu)$ by its sample estimate $\hat{V}_C^{\pi_{\phi_{old}}}(\mu)$ and denote $\chi - \hat{V}_C^{\pi_{\phi_{old}}}(\mu)$ as $\chi'$. Note that $\chi'$ is independent of parameter $\phi$ that is being optimized over. So, the objective becomes

$$\max_{\pi_\phi} \min_{\lambda \geq 0} \mathbb{E}_{s \sim \rho_{\phi_{old}}, a \sim \pi_\phi} \left[ \hat{A}^{\pi_{\phi_{old}}}(s,a) - \frac{\lambda}{1-\gamma} \hat{A}_C(s,a) \right] + \lambda \chi'$$
$$s.t. \quad \mathbb{E}_{s \sim \rho_{\phi_{old}}} [D_{\text{KL}}(\pi_{\phi_{old}}(\cdot|s) || \pi_\phi(\cdot|s))] \leq \delta \tag{44}$$

For notational convenience let $\lambda'$ denote the fraction $\frac{\lambda}{1-\gamma}$. Also, in the expectation, we replace $a \sim \pi_\phi$ by $a \sim \pi_{\phi_{old}}$ and account for it by importance weighting of the objective.

Let us consider $\max_{\pi_\phi}$ operation and the following gradient necessary for gradient ascent of $\phi$

$$\phi \leftarrow \arg\max_\phi \mathbb{E}_{s \sim \rho_{\phi_{old}}} \left[ \mathbb{E}_{a \sim \pi_{\phi_{old}}} \left[ \frac{\pi_\phi(a|s)}{\pi_{\phi_{old}}(a|s)} (A_R^{\pi_{\phi_{old}}}(s,a) - \lambda' \hat{A}_C(s,a)) \right] \right]$$
$$s.t. \quad \mathbb{E}_{s \sim \rho_{\phi_{old}}} [D_{\text{KL}}(\pi_{\phi_{old}}(\cdot|s) || \pi_\phi(\cdot|s))] \leq \delta \tag{45}$$

$$\phi \leftarrow \arg\max_\phi \nabla_{\phi_{old}} \bar{A}(\phi_{old})^T (\phi - \phi_{old})$$
$$s.t. \quad \mathbb{E}_{s \sim \rho_{\phi_{old}}} [D_{\text{KL}}(\pi_{\phi_{old}}(\cdot|s) || \pi_\phi(\cdot|s))] \leq \delta \tag{46}$$

Here, using slide 20 of Lecture 9 in (Levine, 2018), and the identity $\nabla_\phi \pi_\phi = \pi_\phi \nabla_\phi \log \pi_\phi$ we have

$$\nabla_\phi \bar{A}(\phi) = \mathbb{E}_{s \sim \rho_{\phi_{old}}} \left[ \mathbb{E}_{a \sim \pi_{\phi_{old}}} \left[ \frac{\pi_\phi(a|s)}{\pi_{\phi_{old}}(a|s)} \nabla_\phi \log \pi_\phi(a|s) (A_R^{\pi_{\phi_{old}}}(s,a) - \lambda' \hat{A}_C(s,a)) \right] \right] \tag{47}$$

Using slide 24 of Lecture 5 in (Levine, 2018) and estimating locally at $\phi = \phi_{old}$,

$$\nabla_{\phi_{old}} \bar{A}(\phi_{old}) = \mathbb{E}_{s \sim \rho_{\phi_{old}}} \left[ \mathbb{E}_{a \sim \pi_{\phi_{old}}} \left[ \nabla_{\phi_{old}} \log \pi_{\phi_{old}}(a|s) (A_R^{\pi_{\phi_{old}}}(s,a) - \lambda' \hat{A}_C(s,a)) \right] \right] \tag{48}$$

We note that, $\mathbb{E}_{s \sim \rho_{\phi_{old}}} \left[ \mathbb{E}_{a \sim \pi_{\phi_{old}}} \left[ \nabla_{\phi_{old}} \log \pi_{\phi_{old}}(a|s) \hat{A}^{\pi_{\phi_{old}}}(s,a) \right] \right] = \nabla_{\phi_{old}} J(\phi_{old})$, the original policy gradient corresponding to task rewards. So, we can write equation 48 as

$$\nabla_{\phi_{old}} ar A(\phi_{old}) = \nabla_{\phi_{old}} J(\phi_{old}) + \mathbb{E}_{s \sim \rho_{\phi_{old}}} \left[ \mathbb{E}_{a \sim \pi_{\phi_{old}}} \left[ -\lambda' \hat{A}_C(s,a) \right] \right] \tag{49}$$

In practice, we estimate $A_R^{\pi_{\phi_{old}}}$ through GAE (Schulman et al., 2015b;a; Levine, 2018)

$$\hat{A}^{\pi_{\phi_{old}}} = \sum_{t'=t}^\infty (\gamma)^{t'-t} \Delta_{t'} \quad \Delta_{t'} = r(s_{t'}, a_{t'}) + \gamma V_R(s_{t'+1}) - V_R(s_{t'}) \tag{50}$$

Let $\hat{A}^{\pi_{\phi_{old}}}(s,a) = A_R^{\pi_{\phi_{old}}}(s,a) - \lambda' A_C(s,a)$ denote the modified advantage function corresponding to equation 48

$$\hat{A}^{\pi_{\phi_{old}}} = \sum_{t'=t}^\infty (\gamma)^{t'-t} \Delta_{t'} \quad \Delta_{t'} = r(s_{t'}, a_{t'}) + \gamma V_R(s_{t'+1}) - V_R(s_{t'}) - \lambda' \hat{A}_C(s_{t'}, a_{t'}) \tag{51}$$

So, rewriting equations 48 and 53 in terms of $\tilde{A}^{\pi_{\phi_{old}}}$, we have

$$\nabla_{\phi_{old}} \bar{A}(\phi_{old}) = \mathbb{E}_{s \sim \rho_{\phi_{old}}} \left[ \mathbb{E}_{a \sim \pi_{\phi_{old}}} \left[ \nabla_{\phi_{old}} \log \pi_{\phi_{old}}(a|s) \hat{A}^{\pi_{\phi_{old}}} \right] \right] \tag{52}$$

$$\nabla_{\phi_{old}} \bar{A}(\phi_{old}) = \nabla_{\phi_{old}} \tilde{J}(\phi_{old}) \tag{53}$$

Substituting in equation 46, we have

$$\phi \leftarrow \arg\max_\phi \nabla_{\phi_{old}} \tilde{J}(\phi_{old})^T (\phi - \phi_{old})$$
$$s.t. \quad \mathbb{E}_{s \sim \rho_{\phi_{old}}} [D_{\text{KL}}(\pi_{\phi_{old}}(\cdot|s) || \pi_\phi(\cdot|s))] \leq \delta \tag{54}$$

As shown in slide 20 of Lecture 9 (Levine, 2018) and (Schulman et al., 2015a), we can approximate $D_{\text{KL}}$ in terms of the Fisher Information Matrix $\mathbf{F}$ (this is the second order term in the Taylor

expansion of KL; note that around $\phi = \phi_{old}$, both the KL term and its gradient are 0),

$$D_{\text{KL}}(\pi_{\phi_{old}}(\cdot|s)||\pi_\phi(\cdot|s)) = \frac{1}{2}(\phi - \phi_{old})^T \mathbf{F}(\phi - \phi_{old}) \tag{55}$$

Where, $\mathbf{F}$ can be estimated with samples as

$$\mathbf{F} = \mathbb{E}_{s \sim \rho_{\phi_{old}}} \left[ \mathbb{E}_{a \sim \pi_{\phi_{old}}} \left[ \nabla_{\phi_{old}} \log \pi_{\phi_{old}} (\nabla_{\phi_{old}} \log \pi_{\phi_{old}})^T \right] \right] \tag{56}$$

So, finally, we can write the gradient ascent step for $\phi$ as (natural gradient conversion)

$$\phi \leftarrow \phi_{old} + \beta \mathbf{F}^{-1} \nabla_{\phi_{old}} \tilde{J}(\phi_{old}) \quad \beta = \sqrt{\frac{2\delta}{\nabla_{\phi_{old}} \tilde{J}(\phi_{old})^T \mathbf{F} \nabla_{\phi_{old}} \tilde{J}(\phi_{old})}} \tag{57}$$

In practice, we perform backtracking line search to ensure the $D_{\text{KL}}$ constraint satisfaction. So, we have the following update rule

$$\phi \leftarrow \phi_{old} + \beta \boldsymbol{F}^{-1} \nabla_{\phi_{old}} \tilde{J}(\phi_{old}) \quad \beta = \beta^j \sqrt{\frac{2\delta}{\nabla_{\phi_{old}} \tilde{J}(\phi_{old})^T \boldsymbol{F} \nabla_{\phi_{old}} \tilde{J}(\phi_{old})}} \tag{58}$$

After every update, we check if $\bar{D}_{\text{KL}}(\phi||\phi_{old}) \leq \delta$, and if not we decay $\beta^j = \beta^j(1 - \beta^j)^j$, set $j \leftarrow j + 1$ and repeat for $L$ steps until $\bar{D}_{\text{KL}} \leq \delta$ is satisfied. If this is not satisfied after $L$ steps, we backtrack, and do not update $\phi$ i.e. set $\phi \leftarrow \phi_{old}$. For gradient descent with respect to the Lagrange multiplier $\lambda$ we have (from equation 5),

$$\lambda \leftarrow \lambda - \left( \frac{1}{1 - \gamma} \mathbb{E}_{s \sim \rho_{\phi_{old}}, a \sim \pi_{\phi_{old}}} [\hat{A}_C(s, a)] - \chi' \right) \tag{59}$$

Note that in the derivations we have ommitted $\sum_t$ in the outermost loop of all expectations, and subscripts (e.g. $a_t$, $s_t$) in order to avoid clutter in notations.

## A.3 Relation to CPO

The CPO paper (Achiam et al., 2017) considers a very similar overall objective for policy gradient updates, with one major difference. CPO approximates the $V_C^{\pi_\phi}(\mu) \leq \chi$ constraint by replacing $V_C^{\pi_\phi}(\mu)$ with its first order Taylor expansion and enforces the resulting simplified constraint exactly in the dual space. On the other hand, we do not make this simplification, and use primal-dual optimization to optimize an upper bound on $V_C$ through the CQL inspired objective in equation 2. Doing this and not not making the linearity modification allows us to handle sparse (binary) failure indicators from the environment without assuming a continuous safety cost function as done in CPO (Achiam et al., 2017).

## A.4 Practical considerations

Depending on the value of KL-constraint on successive policies $\delta$, the RHS in Theorem 2 can either be a lower or higher rate than the corresponding problem without safety constraint. In particular, let the sampling error $\zeta = 0$, then if $\delta \geq \frac{(1-\gamma)^4(2-\chi)^2}{8\gamma^2}$, the third term is negative.

If we set $\gamma = 0.99$ and $\chi = 0.05$, then for any $\delta > \texttt{1e-8}$, the third term in Theorem 3 will be negative. Also, if $\alpha$ is chosen to be much greater than that in equation 26, the value of $\Delta$ can be arbitrarily increased in principle, and we would be overestimating the value of $Q_C$ significantly. While increasing $\Delta$ significantly will lead to a decrease in the upper bound of $V_R^*(\mu) - V_R^{(T)}(\mu)$, but in practice, we would no longer have a practical algorithm. This is because, when $Q_C$ is overestimated significantly, it would be difficult to guarantee that line 9 of Algorithm 1 is satisfied, and policy execution will stop, resulting in infinite wall clock time for the algorithm.

In order to ensure that the above does not happen, in practice we loop over line 6 of Algorithm 1 for a maximum of 100 iterations. So, in practice the anytime safety constraint satisfaction of Theorem 2 is violated during the early stages of training when the function approximation of $Q_C$ is incorrect. However, as we demonstrate empirically, we are able to ensure the guarantee holds during the majority of the training process.

A.5 DETAILS ABOUT THE ENVIRONMENTS

In each environment, shown in Figure 2, we define a task objective that the agent must achieve and a criteria for *catastrophic failure*. The goal is to solve the task without dying. In all the environments, in addition to the task reward, the agent only receives a binary signal indicatin whether it is dead i.e. a catastrophic failure has occurred (1) or alive (0).

- *Point agent navigation avoiding traps.* Here, a point agent with two independent actuators for turning and moving forward/backward must be controlled in a 2D plane to reach a goal (shown in green in Figure 2) while avoiding traps shown in violet circular regions. The agent has a health counter set to 25 for the episode and it decreases by 1 for every time-step that it resides in a trap. The agent is *alive* when the health counter is positive, and a *catastrophic failure* occurs when the counter strikes 0 and the agent dies.

- *Car agent navigation avoiding traps.* Similar environment as the above but the agent is a Car with more complex dynamics. It has two independently controllable front wheels and free-rolling rear wheel. We adapt this environment from (Ray et al., 2019).

- *Panda push without toppling.* A Franka Emika Panda arm must push a vertically placed block across the table to a goal location without the block toppling over. The workspace dimensions of the table are 20cmx40cm and the dimensions of the block are 5cmx5cmx10cm. The environment is based on Robosuite Zhu et al. (2020) and we use Operational Space Control (OSC) to control the end-effevctor velocities of the robot arm. A *catastrophic failure* is said to occur is the block topples.

- *Panda push within boundary.* A Franka Emika Panda arm must be controlled to push a block across the table to a goal location without the block going outside a rectangular constraint region. *Catastrophic failure* occurs when the block center of mass ($(x, y)$ position) move outside the constraint region on the table with dimensions 15cmx35cm. The dimensions of the block are 5cmx5cmx10cm. The environment is based on Robosuite Zhu et al. (2020) and we use Operational Space Control (OSC) to control the end-effector velocities of the robot arm.

- *Laikago walk without falling*, a Laikago quadruped robot must walk without falling. The agent is rewarded for walking as fast as possible (or trotting) and *failure* occurs when the robot falls. Since this is an *extremely* challenging task, for all the baselines, we initialize the agent's policy with a controller that has been trained to keep the agent standing, while not in motion. The environment is implemented in PyBullet and is based on (Peng et al., 2020).

A.6 HYPER-PARAMETER DETAILS

We chose the learning rate $\eta_Q$ for the safety-critic $Q_C$ to be $2e - 4$ after experimenting with $1e - 4$ and $2e - 4$ and observing slightly better results with the latter. The value of discount factor $\gamma$ is set to the usual default value $0.99$, the learning rate $\eta_\lambda$ of the dual variable $\lambda$ is set to $4e - 2$, the value of $\delta$ for the $D_{\text{KL}}$ constraint on policy updates is set to $0.01$, and the value of $\alpha$ to be $0.5$. We experimented with three different $\alpha$ values $0.05, 0.5, 5$ and found nearly same performance across these three values. For policy updates, the backtracking co-efficient $\beta^{(0)}$ is set to $0.7$ and the max. number of line search iterations $L = 20$. For the *Q-ensembles* baseline, the ensemble size is chosen to be $20$ (as mentioned in the LNT paper), with the rest of the common hyper-parameter values consistent with CSC, for a fair comparison. All results are over four random seeds.

## A.7 COMPLETE RESULTS FOR TRADEOFF BETWEEN SAFETY AND TASK PERFORMANCE

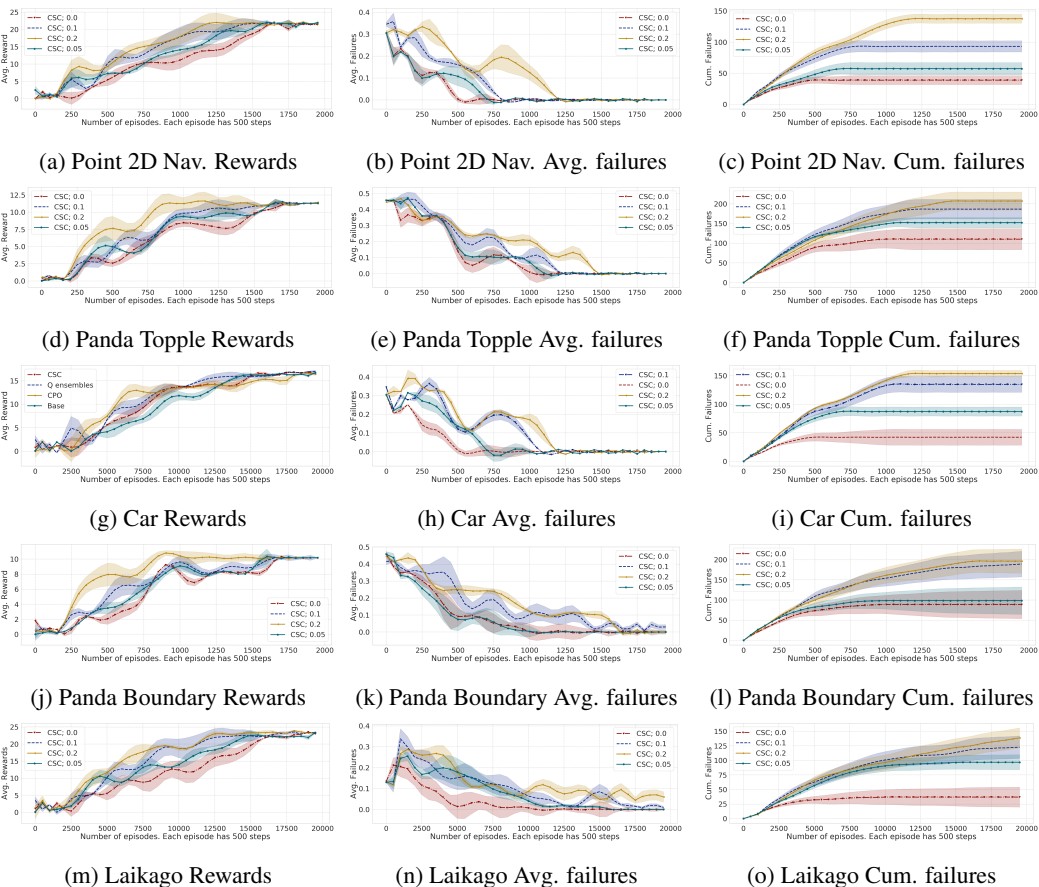

Figure 5: Results on the five environments we consider for our experiments. For each environment we plot the average task reward, the average episodic failures, and the cumulative episodic failures. All the plots are for our method with different safety thresholds $\chi$. From the plots it is evident that our method can naturally trade-off safety for task performance depending on how strict the safety threshold $\chi$ is set to. In particular, for a stricter $\chi$ (i.e. lesser value), the avg. failures decreases, and the task reward plot also has a slower convergence compared to a less strict threshold.

## A.8 Complete results for comparison with baselines

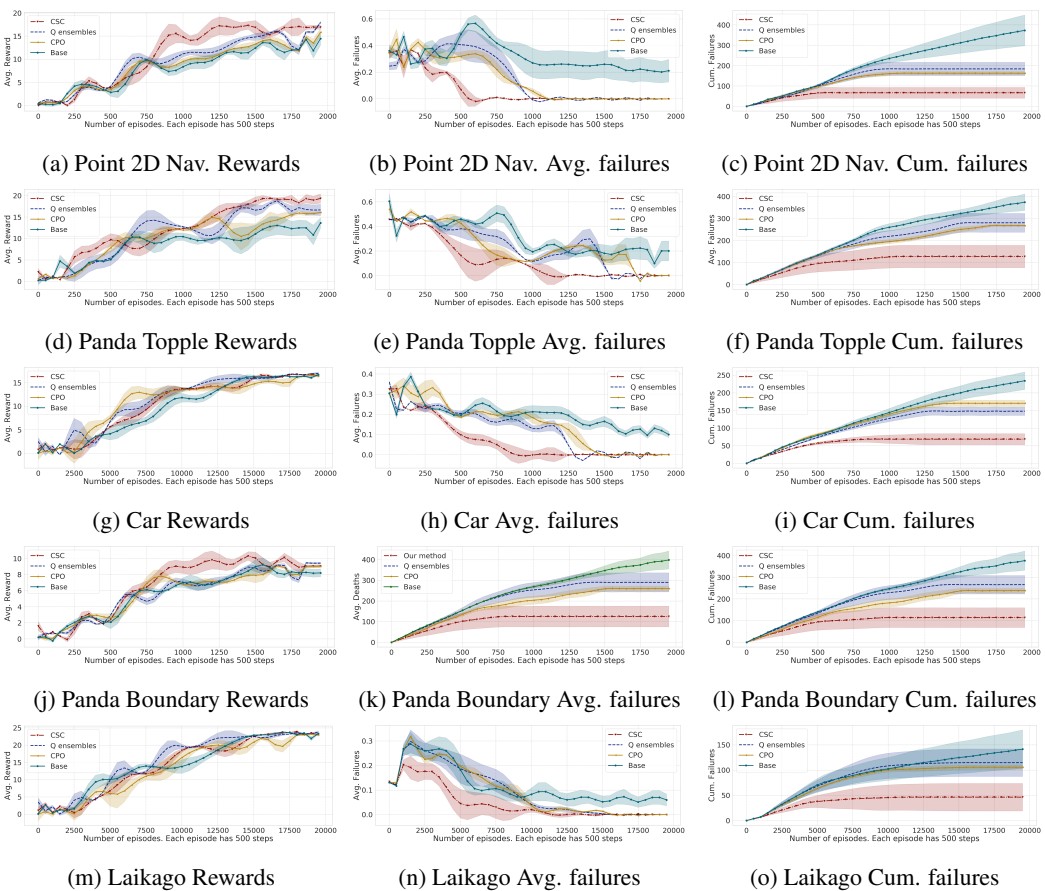

(a) Point 2D Nav. Rewards
(b) Point 2D Nav. Avg. failures
(c) Point 2D Nav. Cum. failures

(d) Panda Topple Rewards
(e) Panda Topple Avg. failures
(f) Panda Topple Cum. failures

(g) Car Rewards
(h) Car Avg. failures
(i) Car Cum. failures

(j) Panda Boundary Rewards
(k) Panda Boundary Avg. failures
(l) Panda Boundary Cum. failures

(m) Laikago Rewards
(n) Laikago Avg. failures
(o) Laikago Cum. failures

Figure 6: Results on the five environments we consider for our experiments. For each environment we plot the average task reward, the average episodic failures, and the cumulative episodic failures. Since Laikago is an *extremely* challenging task, for all the baselines, we initialize the agent's policy with a controller that has been trained to keep the agent standing, while not in motion. The task then is to bootstrap learning so that the agent is able to remain standing while walking as well. The safety threshold $\chi = 0.05$ for all the baselines in all the environments.

## A.9 Comparison between two unconstrained RL algorithms

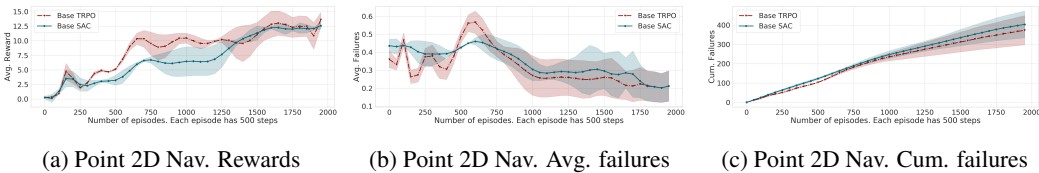

(a) Point 2D Nav. Rewards
(b) Point 2D Nav. Avg. failures
(c) Point 2D Nav. Cum. failures

Figure 7: Comparison between two RL algorithms TRPO (Schulman et al., 2015a), and SAC (Haarnoja et al., 2018) in the Point agent 2D Navigation environment. We see that TRPO has slightly faster convergence in terms of task rewards and also slightly lower average and cumulative failures, and so consider TRPO as the *Base* RL baseline in Figures 3 and 4.

### A.10 SEEDING THE REPLAY BUFFER WITH VERY FEW SAMPLES

In order to investigate if we can leverage some offline user-specified data to lower the number of failures during training even further, we seed the replay buffer of CSC and the baselines with 1000 tuples in the Car navigation environment. The 1000 tuples are marked as *safe* or *unsafe* depending on whether the car is inside a trap location or not in those states. If our method can leverage such manually marked offline data (in small quantity as this marking procedure is not cheap), then we have a more practical method that can be deployed in situations where the cost of visiting an unsafe state is significantly prohibitive. Note that this is different from the setting of offline/batch RL, where the entire training data is assumed to be available offline - in this experimental setting we consider very few tuples (only 1000). Figure 9 shows that our method can successfully leverage this small offline data to bootstrap the learning of the safety critic and significantly lower the average failures. We attribute this to training the safety critic conservatively through CQL, which is an effective method for handling offline data.

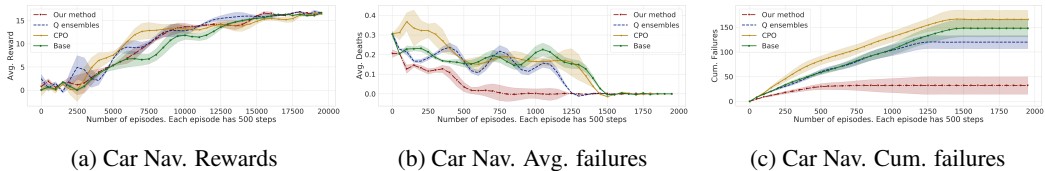

(a) Car Nav. Rewards      (b) Car Nav. Avg. failures      (c) Car Nav. Cum. failures

Figure 8: **Results on the Car navigation environment after seeding the replay buffer with 1000 tuples.** Although all the baselines improve by seeding, in terms of lower failure rates compared to Figure 3, we observe that CSC is able to particularly leverage the offline seeding data and significantly lower the average and cumulative failures during training.

### A.11 CAR NAVIGATION WITH TRAPS USING CONTINUOUS SAFETY SIGNAL

In this section we consider the case of a continuous safety signal to show that CSC can learn constraints in this setting as well, and minimize failures significantly more compared to the baselines. The car navigation with traps provides a natural setting for this, because every time the agent enters a trap region, it can receive a penalty (and its health counter decreases by 1), and a catastrophic failure occurs when the health counter drops to 0. By training the safety critic with this continuous failure signal instead of on binary failure signals, we can capture a notion of *impending failure* and hence aim to be more safe. Note that this setting is a strictly easier evaluation setting than what we previously considered with the binary safety signal.

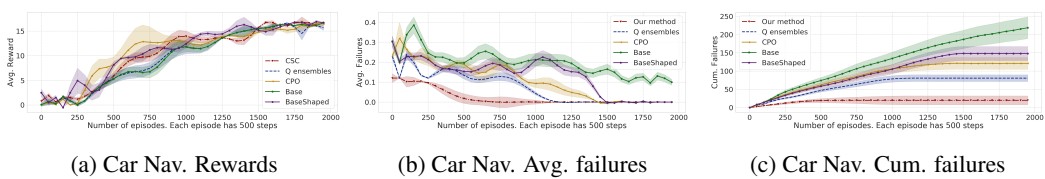

(a) Car Nav. Rewards      (b) Car Nav. Avg. failures      (c) Car Nav. Cum. failures

Figure 9: **Results on the Car navigation environment using continuous safety signal.**

