# OpenReview forum: "Conservative Safety Critics for Exploration"
_ICLR.cc/2021/Conference — ICLR 2021 Poster_

### Official Review · AnonReviewer4 · 2020-10-19
**The paper is generally well written, but lacks of mathematical rigor and makes unverified/ exaggerated claims.**

**Rating:** 6
**Confidence:** 5

**Review:**

In this submission the authors are trying to tackle the very important problem of safe RL with safety guarantees. The problem formulation is rather clear, and the paper is overall well written. The main idea is to formulate the safe RL problem as  a CMDP problem, but with worst-case bounds to ensure that the safety constrained is guaranteed throughout the learning.

The general idea is fine, albeit not new (please check/compare against recent works on robust-CMDPs), however, I do have several issues with the lack of rigor in the mathematical proofs, as well as the rather amplified statements about safety guarantees 'throughout learning'.

Indeed, the CMDP problem (or its worst-case bound) is solved using a Lagrangian formulation, which is well-know as a soft constraints formulation, i.e., you do not have guaranteed safety during learning, as claimed in the Introduction (paragraph 3), and throughout the paper. In that case, the dramatic drone motivational example used in the Introduction (paragraph 1) is and exaggeration, i.e., this method will lead to a crash too.

Besides, mathematically, many of your variables are not defined, not even in the appendix, e.g., in equation (2) $\hat{B}$ is not defined; $\alpha$ which seems a key tuning parameter in Theorem 1 is not defined, etc.

Most importantly, the authors keep referring to the paper about CPO (Achiam et al 17) to support their proofs and technical derivations, whereas that paper is about continuous constraints costs. This paper on the other hand, is clearly using discontinuous constraints, not even $C^{0}$, and thus one cannot just use parts of the results in (Achiam et al 17) without further carful examination of the technical challenges introduced with a discontinuous cost function, e.g., you are using some Taylor developments throughout, while these only make sense for analytic functions, etc.

Furthermore, the probability bounds proposed in Theorem 1 and 2 seem to be rater weak bounds, since $\xi$ is bounded by a term inversely proportional to the confidence parameter $\omega$, which means the probability of being safe is high when the safety bound is loose, i.e., $\chi+\xi$, for $\xi\rightarrow +\infty$ . Similarly, for tight safety bound, i.e., $\chi+\xi$, for $\xi\rightarrow 0$, the safety probability drops to zero. The authors seem to minimize this point by the 'proper tuning' of $\alpha$ which remains a mysterious parameter (even after checking the proofs in Appendix).

This is all to say that the authors have to tone down their statements about guaranteed provable safety bound, quite a bit.

Finally, the numerical simulations are interesting, but only confirm my point about the fact that the obtained safety is asymptotic only, i.e., in steady state, and absolutely not during learning. This is clear from the plots in Figure 3 (bottom) where we see that in average the proposed algorithm necessitates more than 1000 iteration before reaching a truly safe behavior. One can also note that the CPO algorithm behaves almost similarly to the proposed algorithm, when tested on the car navigation example, and the laikago robot (Fig. 3, bottom number 3 and 4). One also wonders why in the first set of tests, in Fig. 3- top number 2, one can reach a performance cost of 20 with the proposed method, while it seems to plato at 10 in the second set of experiments, in Fig. 4-top number 2. Another point that is worth clarifying is that the tests in Fig. 4-top number 1, we see that the performance cost reaches 10 for the proposed method with large safety constraint bound (0.2), which is intuitively an almost unconstrained case. However, the unconstrained algorithm 'Base' in the first tests, Fig. 3-top number 1, the Base algorithm does not achieve a similar performance, could it be better tuned in that case ?

In summary, I found the paper well written, but it needs to be carefully revised for technical rigor, and toned down in terms of what is really achieved here (maybe somehow safer CMDP algorithm but definitely not safe during exploration).

---

> ### Author Response · Authors · 2020-11-16
> **Author response (part 2/2) : Response to other comments**
>
> **Author response (part 2/2) : Response to other comments**
>
> **In that case, the dramatic drone motivational example used in the Introduction (paragraph 1) is and exaggeration, i.e., this method will lead to a crash too.**
>
> We have now replaced this example in the revised paper. Thank you for pointing this out.
>
>
> **Besides, mathematically, many of your variables are not defined, not even in the appendix, e.g., $\hat{B}^{\pi_\phi}$ in equation (2)  is not defined;  $\alpha$ which seems a key tuning parameter in Theorem 1 is not defined, etc.**
>
> Thank you for spotting this - we have defined the Bellman operator $\hat{B}^{\pi_\phi}$ in the revised paper. Kindly note that we have the minimum theoretically necessary value of $\alpha$ defined in equation 27 of the Appendix. We have now included a line about $\alpha$ being a parameter that weighs the first term in equation (2). It is inherited from the CQL paper (Kumar et al. 2020).
>
> **The authors seem to minimize this point by the 'proper tuning' of  which remains a mysterious parameter (even after checking the proofs in Appendix).**
>
> We request you to kindly refer equation (27) in the Appendix where we had derived the minimum theoretically specified value of $\alpha$ for obtaining a tight bound.
>
> **Finally, the numerical simulations are interesting, but only confirm my point about the fact that the obtained safety is asymptotic only, i.e., in steady state, and absolutely not during learning. This is clear from the plots in Figure 3 (bottom) where we see that in average the proposed algorithm necessitates more than 1000 iteration before reaching a truly safe behavior.**
>
> Yes, we agree with you that it is not possible to obtain a good, near-optimal policy without hitting some failures during training. The amount of safety violation during training depends on the need of the task, and is not an absolute number. We kindly refer you to the fact that even during training, after around 500 iterations, the avg. failure rate drops to about $10\%$ for our method CSC, and becomes almost $0$ after about 1000 iterations. We do not claim to have $0$ failures during training, but have low average failures.
>
>
> **One can also note that the CPO algorithm behaves almost similarly to the proposed algorithm, when tested on the car navigation example, and the laikago robot (Fig. 3, bottom number 3 and 4).**
>
> Please note that there is a large gap (the gap is more than Mean$\pm$S.D) between the average failures of CSC and CPO in the car Navigation and Laikago environment, although eventually CPO also achieves nearly $0$ average failures, which is expected since all the methods succeed in satisfying constraints at convergence. The differences are pronounced in the early stages of training. We kindly request you to also look at the plots of total cumulative failures in section A.8 of the Appendix where the differences are more clearly seen - CSC has around $50$ failures during the entire training process, while CPO has $>100$.
>
> **One also wonders why in the first set of tests, in Fig. 3- top number 2, one can reach a performance cost of 20 with the proposed method, while it seems to plato at 10 in the second set of experiments, in Fig. 4-top number 2.**
>
> Kindly note that the reward functions for the two environments are defined differently, and the size of the workspace are different (Section A.5 of the Appendix), so the absolute numbers on y-axis for the two plots cannot be directly compared.

---

> ### Author Response · Authors · 2020-11-16
> **Author response (part 1/2) : Summary, clarification about safety bounds, clarification about continuous constraints**
>
> Thank you for the detailed review and comments about our paper. We have removed the phrase *safety guarantee* from the paper and replaced it with what we actually show through the theoretical and empirical results: upper bound on the probability of safety violations (failures).
>
> **Added result to show cumulative number of safety violations decreases quickly:** We have also added a new theoretical result (*Theorem 3*) that shows that by setting the threshold for safety violation, \chi_t, in a way that it decreases over time,  the *cumulative* number of failures grows only sublinearly with respect to the amount of data collected. This means that the number of failures quickly goes down to 0, at a rate of $\mathcal{O}(\sqrt{T})$. While no algorithm can explore alongside being safe every time (i.e. incurring absolutely no failure during learning) without knowing the safety constraints beforehand, we now show that our method reduces the probability of failures at a fast rate as a function of how much data is collected, implying that it leads to a smaller number of failures faster.
>
> **Addressed discontinuous constraints:**  We have also responded to the comment about discontinuous constraint, by pointing out that the constraint we have in terms of $V_C$ (in equation 2) is indeed continuous, as it is defined as an expectation over episodic failures. So, although $C(s)$ is binary indicating if a failure has occurred, the constraint is in terms of  $V_C$ ($V_C^\pi(\mu) = E_{\tau\sim\pi}\left[\sum_{t=0}^{\infty}C(s_t)\right]\leq\chi$), which is continuous.
>
> In the points below, we discuss the main concerns of the reviewer in bold, and provide our responses in plaintext.
>
> **Indeed, the CMDP problem (or its worst-case bound) is solved using a Lagrangian formulation, which is well-know as a soft constraints formulation, i.e., you do not have guaranteed safety during learning, as claimed in the Introduction (paragraph 3), and throughout the paper.**
>
> We respectfully clarify that through safety bounds in the paper, we have not tried to claim $0$ failures during learning (which is not possible to achieve when learning purely from data collected online). We have removed the word \textit{guarantee} in the updated paper to accurately reflect this. We would like to emphasize that although a Lagrangian formulation does not ensure constraint satisfaction during learning, we have proved that our method enforces a (probabilistic) constraint at each point in training, not just in the end. This is the key result in Theorem 1 of the paper. In addition, Theorem 3 shows that training with Algorithm 1 will converge to a “safe” policy that incurs no
> failures at a quick rate, that is atmost sublinear in the number of  environment interactions so far. We verify this empirically, and through the results (Fig. 3) we see that the number of failures are bounded during training, and drops quickly to less than $10\%$ after 1000 iterations. As we mention in section 7, it is not possible to have $0$ failures due to function approximation error in the early stages of training.
>
>
> **Most importantly, the authors keep referring to the paper about CPO (Achiam et al 17) to support their proofs and technical derivations, whereas that paper is about continuous constraints costs. This paper on the other hand, is clearly using discontinuous constraints, not even , and thus one cannot just use parts of the results in (Achiam et al 17) without further carful examination of the technical challenges introduced with a discontinuous cost function, e.g., you are using some Taylor developments throughout, while these only make sense for analytic functions, etc.**
>
> Kindly note that although $C(s)$ outputs binary values depending on whether an accident has occurred in state $s$, the constraint as shown in equations (1) and (3) of the paper are on $V_C$, which is indeed a continuous function of the policy $\pi$. This is akin to a value function for sparse rewards (that takes on 0 or 1) in standard RL. The value function is an expectation of episodic returns and hence is continuous.  $V_C^\pi(\mu) = E_{\tau\sim\pi}\left[\sum_{t=0}^{\infty}C(s_t)\right]$. For example if $n$ trajectories $\tau$ are sampled, each trajectory can have $\sum_{t=0}^{\infty}C(s_t)$ equal to a $1$ (failure; episode terminates) or $0$ (no failure at any step), and so $V_C$ in this case is an average of $n$ numbers, each either $0$ or $1$. The derivative of $V_C$ with respect to the stochastic policy exists and can be computed similar to policy gradient. We kindly request you to re-consider our analysis based on this clarification - we believe that our analysis based on CPO holds since the constraint $V_C$ (which we use for Taylor analysis) is continuous and hence analytic.

---

> > ### Comment · AnonReviewer4 · 2020-11-20
> > **Thank you for the effort in revising the paper. Unfortunately, I still do not understand the argument of continuity of the function series, let alone its analyticity !**
> >
> > First, thank you for the revision effort. I appreciate that the authors have revised the tone of the paper and made clearer the notion of safety and what is actually achieved here.
> >
> > On the technical side, I am not trying to be harsh, but I simply still don't see why you get to the conclusion that your function series converges (pointwise or uniformly) to a continuous function ? I think this needs to be considered carefully, or at least the authors should cite a mathematical report to back up this claim. Besides, sorry to add but continuity does not imply analyticity in the context of function series (e.g., see the results on differentiability in https://www.math.ucdavis.edu/~hunter/m125a/intro_analysis_ch5.pdf), nor in the context of a simple function, e..g., $f(x)=e^{-1/x},\;x>0$, and $f(x)=0\;\text{elsewhere}$, is continuous but not analytical,  on any interval containing  0 in its interior, as the Taylor expansion would lead to the zero function.
> >
> > This being said, I like the revised version of the paper better than the original, because it simply shows what is achieved as well as what is not achieved, i.e., the transient safety during learning, which is a very hard problem indeed. Based on this I am willing to revise my ranking up.

---

> > > ### Author Response · Authors · 2020-11-21
> > > **Thank you for the revised assessment of our paper. We address the point about differentiability of the safety value function with respect to the policy below**
> > >
> > > We thank the reviewer for their prompt response. We are glad that the concerns regarding the notion of safety are addressed.
> > >
> > > First of all, we would like to thank the reviewer for sharing the reference.. We apologize for implying that continuity is sufficient for a function to be analytic.
> > >
> > > As a reference for differentiating the value function with respect to the policy parameters, we would like to draw the reviewer’s attention to the policy gradient theorem paper (Sutton et al. 2000 [1]), where it is shown that the gradient $\frac{\partial V^\pi}{\partial \theta}$ exists and can be computed for all parameteric stochastic policies. Here, $V$ denotes the  expected value function of a parameteric policy $\pi_\theta$ under the initial state distribution, which is analogous to the function $V^\pi_C$ in Equation 1 (and section 2) of our paper. The rewards $r_t$ at every time-step defined in Section 1 of [1] are any real numbers, in particular can also be a discontinuous function  or non-differentiable, such as a binary-valued function...
> > >
> > > We want to emphasize that the function  $V_C^\pi$ ( Equation 1 and section 2) in our paper is formulated *exactly* as $V^\pi$ in the previous paragraph, with $r$ being replaced by $C$ that takes on value 0 or 1. In addition, our parameteric policy is modelled as a Gaussian distribution, which is non-deterministic. So, we believe that as a consequence of the  existence of policy gradients  which is equal to the derivative of the value function $V$ with respect to $\pi$, $V_C^\pi$ should also be differentiable with respect to $\pi$ and the Taylor theorem based derivations would follow.
> > >
> > > In addition, while $C(s)$ is a discontinuous function of the state $s$, the derivative required in arriving at our proofs based on CPO is the derivative of the value function $V^\pi_C$ with respect to the policy $\pi$. Please let us know if this response clarifies the point better. We would be happy to provide additional clarification, as well as add this in the paper. We apologize if we are misunderstanding the reviewer’s question.
> > >
> > >
> > >
> > > Thank you so much for suggesting to revise our ranking up. We would appreciate it if the reviewer can update the score to reflect this positive assessment of the paper.
> > >
> > >
> > > [1] Sutton, R. S., McAllester, D. A., Singh, S. P., & Mansour, Y. (2000). Policy gradient methods for reinforcement learning with function approximation. In Advances in neural information processing systems (pp. 1057-1063).

---

> > > > ### Comment · AnonReviewer4 · 2020-11-24
> > > > **Thank you for the clarification, and for adding reference [1] in your response.**
> > > >
> > > > I will certainly read this reference with interest; I hope that the authors can add some discussion about the analyticity of $V^{\pi}_{C}$, perhaps by explicitly referring to [1]. The reason is that some people, like myself, who are more applied mathematicians than RL specialist, will have the opportunity to think about this point more carefully when reading your paper, and perhaps even challenge the results in [1] if there are some technical details that have been overlooked (I think you can guess that I am still a bit skeptical). Regardless, I think that with these changes reflected in the paper, the final version of the paper has his own merits, including what needs to be clarified, and my ranking has been updated up, from 5 to 6.

---

> ### Author Response · Authors · 2020-11-20
> **Discussion**
>
> Kindly let us know if our response below addressed your concerns. We will be happy to answer if there are additional issues/questions.

---

### Official Review · AnonReviewer3 · 2020-10-28
**Theoretical and practical advances in safe exploration**

**Rating:** 7
**Confidence:** 2

**Review:**

(some difficulty following all of the proofs lowered the confidence of my evaluation)

## Summary
The authors lay out a new technique for safety-constrained exploration which reduces the likelihood of catastrophic failure. This is shown via extensive proofs providing theoretical guarantees on both convergence and the likelihood of failure as well as experimental results in a number of compelling tasks.

## Quality & Clarity
The authors provide regular and clear comparisons of their approach to related techniques described in other works. They provide extensive proofs and experimental results justifying their technique’s advantages.

## Originality & Significance
The work provides original techniques with both theoretical and empirical improvements over previous techniques.

## Suggestions
In the experimental section, the positive reward is clear but what is missing is how receiving negative reward for failure might compare to this approach. A natural comparison against methods that do not utilize the safety constraint features is to include a very large negative reward for failure and see if the algorithms can avoid failure via this signal instead of the separate safety signal. The authors emphasize the value of only accessing a binary signal to signify safety from the environment, so this could easily be considered a slight modification of the baseline algorithms: they induce a large negative intrinsic reward when they receive a failure signal. The trap experiment provides an especially clear setting for negative rewards: the agents could receive a lesser negative reward each time they lose a health counter before reaching catastrophic failure.

During policy evaluation, you resample actions when the likelihood of failure for a given action exceeds a certain threshold. I wonder how alternative approaches might perform, such as using the failure likelihood instead as a probability (perhaps scaled) of resampling; reweighting the action probabilities based on the failure likelihoods (maybe another action had a nearly identical expected return but a much lower chance of failure); or even utilizing the failure likelihood as an input to the policy (maybe in some scenarios the model stays as far away as possible from any risk but in others might choose to take a dangerous route that could lead to high reward). These alternatives also provide an answer to “what do you do when every action from a given state is above the threshold?” which is not specified in Section 3’s “Executing rollouts (i.e., safe exploration)” subsection nor in Algorithm 1.

Alternative approaches to the sampling procedure could potentially improve the tradeoff between safety and convergence. Where other methods might waste time exploring dangerous parts of the space, your method might be able to focus more time on the healthier parts of the states and converge to an effective policy sooner. At least, this may be true after the initial period of learning an accurate safety classifier, which as you point out takes some time for the agent to figure out. An experiment that could add to the paper would be to pretrain a safety estimator and then restart training with a newly initialized policy--how does this affect the convergence with different thresholds?

Finally, I would be interested to see how this work could be extended with non-binary safety constraints. The trap task provides a clear example of incremental danger (the agent is being “injured” before being finally destroyed) and being able to effectively utilize earlier hints signalling impending catastrophic failure would be a valuable skill.

---

> ### Author Response · Authors · 2020-11-16
> **Author response: Baseline with negative rewards, offline data to train safety estimator, evaluation on a non-binary safety indicator setting**
>
> Thank you for the detailed review of our paper and the encouraging comments. The main comments in the review were about additional experiments to better understand the approach, which we have addressed in the responses and below, and in the revised paper. In particular, we have **added a new experiment with a baseline "BaseShaped"** modified from the unconstrained baseline "Base" that incorporates safety violations as negative rewards, as proposed by the reviewer, clarified an existing result with offline data for pre-training the safety estimator, and  added results for continuous safety indicator in the car navigation with traps environment.  We have also provided more intuitions for the theorems and proofs.
>
> Please find our detailed responses pointwise below, with reviewer comments in bold and our replies in plaintext:
>
> **In the experimental section, the positive reward is clear but what is missing is how receiving negative reward for failure might compare to this approach. A natural comparison against methods that do not utilize the safety constraint features is to include a very large negative reward for failure and see if the algorithms can avoid failure via this signal instead of the separate safety signal. The authors emphasize the value of only accessing a binary signal to signify safety from the environment, so this could easily be considered a slight modification of the baseline algorithms: they induce a large negative intrinsic reward when they receive a failure signal.** Thank you for suggesting this baseline. We have implemented this now, and added the results in the revised paper for Fig. 3. This new baseline is named "BaseShaped" and it performs better than "Base" in terms of minimizing the number of failures. It also achieves 0 failures asymptotically, unlike Base.
>
> **An experiment that could add to the paper would be to pretrain a safety estimator and then restart training with a newly initialized policy--how does this affect the convergence with different thresholds?** Thank you for suggesting this. Kindly note that we had experimented with a very similar setting by seeding the replay buffer with some offline data for training the critic (section A.10). The main issue with a completely offline pre-training of the critic (the safety estimator) is how to collect the data in the first place without being unsafe. For the car navigation with traps environment in section A.10 Figure 8 , we consider a "small" offline dataset where we manually marked 1000 states as being safe/unsafe, but in the general case and with a large dataset, this is intractable. We are happy to experiment with any other suggested scheme that the reviewer might suggest.
>
> **Finally, I would be interested to see how this work could be extended with non-binary safety constraints. The trap task provides a clear example of incremental danger (the agent is being “injured” before being finally destroyed) and being able to effectively utilize earlier hints signalling impending catastrophic failure would be a valuable skill.** Yes, we had results for this non-binary safety indicator in the trap task, which we did not include in the paper to avoid confusion. We have now added these plots in section A.12 Figure 9. We believe that this setting of a continuous safety indicator requires more domain knowledge to design a shaped function indicating impending danger, but where available easily - for example in the trap task - should be used for training the safety critic. We hope that the results for non-binary safety indicator shows that our method is applicable to this setting as well.

---

### Official Review · AnonReviewer2 · 2020-10-29
**This paper introduces a method for performing safe exploration in RL. It addresses the problem of ensuring that partially-trained policies do not visit unsafe regions of the state space, while still being exploratory enough to collect useful training experiences. This method makes fewer assumptions than those required by related techniques. The experiments are well-designed and convincing, and include evaluations of the proposed method (CSC) in 4 different domains.**

**Rating:** 7
**Confidence:** 3

**Review:**

This paper introduces a method for performing safe exploration in RL. It addresses the problem of ensuring that partially-trained policies do not visit unsafe regions of the state space, while still being exploratory enough to collect useful training experiences. The proposed technique is based on learning conservative estimates of the probability of a catastrophic failure occurring at different states. Based on these, the authors show that it is possible to upper bound the likelihood of reaching an unsafe state at every training step, thereby guaranteeing that all safety constraints are satisfied with high probability. Importantly, the authors also show that (at least asymptotically), the method is no worse than standard unsafe reinforcement learning algorithms.

Overall, this is a well-written paper with sound mathematical arguments. The authors present a thorough review of related work, such as constrained MDPs, and argue that the proposed method requires fewer assumptions. For example, the proposed technique assumes access only to a sparse (binary) indicator of whether entering a particular state would result in catastrophic failure. The authors formally show that it is possible to upper bound the expected probability of failure during every policy update iteration (Thm1), which is a non-trivial result. All update equations are carefully derived and discussed in the appendix. The experiments are well-designed and convincing, and include evaluations of the proposed method (CSC) in 4 different domains, including an 18-DoF quadruped robot.

I believe that this paper introduces an important contribution to the RL community that is concerned with safety. It presents a principled method and introduces non-trivial bounds. In my opinion, this conference's community would benefit from having this paper accepted to its proceedings.


I have just a few questions for the authors:

1) safe exploration in RL is often expressed in terms of ensuring that, after every policy update, the new policy is no worse than the current one (or is worse by a bounded amount). In your paper, by contrast, safe exploration is related to the ability to avoid particular states. Can any of the bounds derived in this paper be used to ensure the former type of safety?

2) during policy evaluation, the proposed algorithm uses the safety critic, Q_C(s,.), to estimate how unsafe a particular state is. If an action sampled by the policy is deemed unsafe, the algorithm repeatedly re-samples new actions. Is this type of rejection sampling guaranteed to always stop? What happens if all actions available in s are unsafe? What if safe actions do exist but have zero probability under the current policy? Could the algorithm get stuck and stop exploring?

3) in Eq2, what is \hat{Beta}^pi?

4) the bound presented in Theorem 2 depends on an additional term K, which (so you argue) can be made small by picking alpha appropriately. How do you pick alpha, in practice, so that the algorithm trades-off convergence rate and safety?

5) when deriving the upper bound on V_C^pi_new (Eq26), you point out the epsilon_C cannot be computed exactly before the update, since it depends on the new optimized policy pi_new. How loose does the upper bound on V get by placing a trivial upper bound epsilon_C (epsilon_C <= 2)? Also, given that policy updates are constrained to keep the stationary distribution of successive policies similar, could this be exploited to achieve a tighter bound on epsilon_C?

---

> ### Author Response · Authors · 2020-11-16
> **Author response: safe exploration clarification, rejection sampling details, trading off safety and performance**
>
> Thank you for the very detailed review of our paper and the encouraging comments about the approach and results. We have responded to your questions pointwise below:
>
> - **safe exploration in RL is often expressed in terms of ensuring that, after every policy update, the new policy is no worse than the current one (or is worse by a bounded amount). In your paper, by contrast, safe exploration is related to the ability to avoid particular states. Can any of the bounds derived in this paper be used to ensure the former type of safety?** The notion of safe exploration that the reviewer is referring to, also referred to as safe policy improvement, is certainly an interesting one. The bounds derived in this paper pertain to a different notion of safety violations -- i.e, the agent satisfies an upper bound on the probability of failure for an environment-specified constraint, while safe policy improvement pertains to a monotonic policy improvement trend. These two aspects are orthogonal and our analysis tools can potentially be used for safe policy improvement as well.
>
> - **during policy evaluation, the proposed algorithm uses the safety critic, $Q_C(s,.)$, to estimate how unsafe a particular state is. If an action sampled by the policy is deemed unsafe, the algorithm repeatedly re-samples new actions. Is this type of rejection sampling guaranteed to always stop? What happens if all actions available in s are unsafe? What if safe actions do exist but have zero probability under the current policy? Could the algorithm get stuck and stop exploring?** Thank you for pointing this out. We discuss this in Section A.4 of the Appendix. It can indeed happen that the rejection sampling procedure does not yield an action with a non-zero probability, especially in the early stages of training. So, in order to avoid getting stuck completely, we loop over for a maximum of 100 iterations. Kindly note that if no action is found such that $Q(s,a) < \epsilon$ is ensured, it does not necessarily mean that all actions in state s, under the current policy will yield a failure. It could be that the learned critic is not completely trained. In this case, we just choose the action for which $Q(s,a)$ is minimum
>
> - **in Eq2, what is $\hat{B}^\pi$?** Thank you for noticing this. We missed explaining this term in the submission. That is the Bellman operator. We have explained it in the revised manuscript.
>
> - **the bound presented in Theorem 2 depends on an additional term K, which (so you argue) can be made small by picking $\alpha$ appropriately. How do you pick $\alpha$, in practice, so that the algorithm trades-off convergence rate and safety?** Yes, K can be made small by choosing alpha appropriately as shown in equation 27 of the Appendix. We have the term $\alpha$ from the CQL bound. We have discussed the value of $\alpha$ in Section A.6 of the Appendix.

---

### Official Review · AnonReviewer1 · 2020-10-30
**Review1**

**Rating:** 6
**Confidence:** 3

**Review:**

This paper would like to address the problem of ``"safe exploration" with a conservative estimation of the environment. Although the problem seems reasonable, I have the following several concerns on this paper:

- Will the safety constraints be revealed to the agent? Standard RL assumes the reward is not revealed to the agent, and I feel the safety constraints cannot be revealed to the agent in prior as well.

- If the safety constraints is not revealed to the agent, then how to train the safety critic? In my opinion, when the agent collect the data that fail catastrophically, then the exploration is already not safe.

- I feel the derivation part in Section 3 is not so clear, as several notations have been introduced without explanation, though it is not hard to understand.

- I don’t understand what Lemma 1 and Theorem 1 can indicate in practice. When we estimate V_C, we already satisfies several catastrophic failures, otherwise we will know nothing about the possible failure. This can also be seen from the traditional exploration analysis. And traditional exploration analysis also says when we have sufficient number of data, the probability that we suffer from catastrophic failure will decrease. In one word, I don’t see any significant idea of ``safe exploration’’ from Lemma 1 and Theorem 1.

- Also, from the traditional exploration analysis, we have the regret lower bound, which says we must suffer from failure to get the best policy. We also have algorithms to match these regret lower bound on some simplified MDP settings, and even stronger guarantee that with high probability we will make small mistake when we have sufficient samples (see the mistake version of IPOC bound in [1]). I don’t feel the current paper present such kinds of guarantee.

A summarize of point 4 & 5: to me I don’t feel the authors well-defined the `''safe exploration''  and give a rigorous analysis on the ``the safety guarantee. For me, I prefer to define the "safe exploration" as minimizing the total failure when finding the optimal policy, but no matter how to define it,  I feel what the authors have done is not "safe".

I feel Theorem 2 is a straightforward adaption of the existing results in [2]. Though seems correct, the authors argue that Theorem 2 shows a tradeoff between safety and convergence, which I cannot agree. If we want the policy to converge to the global optimum, then we need to collect samples from the dangerous region (otherwise we cannot know if that’s dangerous and probably we can get some improvement at that region), how to keep "safe" at that time? Also, a slow convergence will need to run for a longer time and thus need more samples, which can enlarge the probability of catastrophic failure, right?

Overall, though the empirical performance on some benchmark shows the proposed method is better than some of the previous methods, I don’t think the proposed methods really address the issue of the safe exploration. The authors does not formally define what is the safe exploration (and even little about exploration in fact), and the theoretical analysis also doesn’t address any of the safety issues. From my point of view, to achieve good performance, one must suffer from some failures, and our goal should be minimizing the number of failures, rather than minimizing the failure probability of each turn. Unfortunately, the authors does not characterize all of these issues I care about. I would like to say, considering the unsatisfactory of the problem description and theoretical analysis, I don’t think this paper is suitable for publication.

[1] Dann, Christoph, et al. "Policy certificates: Towards accountable reinforcement learning." International Conference on Machine Learning. PMLR, 2019.

[2] Agarwal, Alekh, et al. "Optimality and approximation with policy gradient methods in markov decision processes." arXiv preprint arXiv:1908.00261 (2019).

------------------
The authors answers one of the most important concerns, I have raised score to 6.

---

> ### Author Response · Authors · 2020-11-16
> **Author response (part 3/3): clarification about Theorem 2, references**
>
> due to space constraint, we are creating another comment.
>
> **Though seems correct, the authors argue that Theorem 2 shows a tradeoff between safety and convergence, which I cannot agree. If we want the policy to converge to the global optimum, then we need to collect samples from the dangerous region (otherwise we cannot know if that’s dangerous and probably we can get some improvement at that region), how to keep "safe" at that time? Also, a slow convergence will need to run for a longer time and thus need more samples, which can enlarge the probability of catastrophic failure, right?**
>
> Since we bound the probability of failure after every policy update (Theorem 1), the total failures until convergence is also bounded. In contrast, although the unconstrained problem might converge to the optimal policy with respect to task performance faster, it incurs significantly more failures as there is no bound on the probability of failure during training.The slower convergence rate of the task value function (Theorem 2 relates to how quickly the policy converges to the optimal policy in terms of task performance) is in relation to the unconstrained RL algorithm that does not take safety constraints into account during training. We also verify this empirically in Fig 3 where we see that although the unconstrained RL algorithm (Base) converges quickly in terms of task reward, it incurs more failures.
>
>
> [1] Altman, Eitan. Constrained Markov decision processes. Vol. 7. CRC Press, 1999.
>
> [2] Achiam, J., Held, D., Tamar, A. and Abbeel, P., 2017. Constrained policy optimization. arXiv preprint arXiv:1705.10528.
>
> [3] Wachi, A. and Sui, Y., 2020. Safe reinforcement learning in constrained markov decision processes. arXiv preprint arXiv:2008.06626.
>
> [4] Chow, Y., Nachum, O., Faust, A., Duenez-Guzman, E., and Ghavamzadeh, M. Lyapunov-based safe policy optimization for continuous control. arXiv preprint arXiv:1901.10031, 2019.
>
> [5] Garcıa, J. and Fernández, F., 2015. A comprehensive survey on safe reinforcement learning. Journal of Machine Learning Research, 16(1), pp.1437-1480.

---

> ### Author Response · Authors · 2020-11-16
> **Author response (part 2/3): Safety constraints, Theorem 1 clarification, New regret bound (Theorem 3)**
>
> **How are safety constraints handled - are they revealed to the agent? If they are not, then how to train the safety critic?**
>
> The constraint is not revealed to the agent except through samples (i.e., if the agent violates the constraint, it is told that it violated the constraint). As we mention in equations 1 and 3 of the paper, the constraint is on $V_C^\pi(\mu) = E_{\tau\sim\pi}\left[\sum_{t=0}^{\infty}C(s_t)\right]$. The agent only has access to the value of $C(s_t)$ denoting whether a failure has occurred in state $s_t$ (the agent does not have access to the functional form of $C$ itself), and the value of the task reward $R(s_t,a_t)$. $C(s_t)$ is very similar to a sparse reward signal in standard RL.
>
> It is indeed true that in purely online training with our method, the agent *must fail a few times to collect data of failure states for training the safety critic*.  We attempted to explain the following nuance in Sec. 7, and will prioritize it earlier in the paper: although we cannot ensure $0$ failures in the initial stages of training, we show in Figs. 3 and 4, that we can bring down the average failures to a very low number. Theoretically, through a new result in Theorem 3, we have shown that the number of failures falls off rapidly as training progresses.
>
>
>
> **What Lemma 1 and Theorem 1 can indicate in practice? In one word, I don’t see any significant idea of "safe exploration" from Lemma 1 and Theorem 1.**
>
> Kindly note that in this paper, we used the framework of a Constrained MDP (CMDP), where the objective of safe exploration is to respect the (safety) constraints that are specified, while searching for the optimal policy in terms of task reward (equation 2 in our paper). The safety constraint $V_C^\pi(\mu) \leq \chi$  that we have defined as part of the CMDP is to ensure the expected probability of failures in an episode is bounded by $\chi$. Theorem 1  shows that, after every policy update, we are able to satisfy $V_C^{\pi_{\phi_{new}}}(\mu) \leq \chi$ with a high probability.  - this directly relates to safe exploration in a CMDP because it quantifies a bound on the probability of failing (i..e being unsafe) while exploring.  Kindly note that since Theorem 1 shows that the expected probability of failure is bounded for every policy update iteration, the number of failures is also bounded (Number of failures in $n$ episodes = Probability of failure in each episode x $n$)
>
>
>
>
> **Also, from the traditional exploration analysis, we have the regret lower bound, which says we must suffer from failure to get the best policy. We also have algorithms to match these regret lower bound on some simplified MDP settings, and even stronger guarantee that with high probability we will make small mistake when we have sufficient samples (see the mistake version of IPOC bound in [1]). I don’t feel the current paper present such kinds of guarantee.**
>
>
> Thank you for this insight. We have now added a new theorem, Theorem 3 in the revised paper  that shows that the cumulative failures until a certain number of samples T of the algorithm grows sublinearly when executing Algorithm 1, provided the safety threshold $\chi$ is set to decrease as more episodes are taken (sub-linear regret in terms of failures). Since we can control the safety threshold during optimization, we can guarantee a sublinear growth in the number of failures as more training is performed, indicating that our method quickly decreases the probability of failure, similar to standard exploration analysis where sublinear regret algorithms are considered good.

---

> ### Author Response · Authors · 2020-11-16
> **Author response (part 1/3): Summary of our response**
>
> Thank you for the detailed review of our paper. We believe the main reservations for the review are about how safety constraints are handled in the algorithm, what exactly are we demonstrating with respect to bounds on failures through the theoretical and empirical results, and how is the proposed algorithm doing `` "safe exploration". We address each of these points as: the constraint is not revealed to the agent except through samples (i.e., when the agent violates the constraint, it is told that it violated the constraint and it doesn’t have any other access to the constraint function), our results provide bounds on the probability of failure in each data-collection round, which translates directly to minimizing the number of failures during exploration, and so the proposed algorithm satisfies the constraints of the CMDP [3] , which is the framework we use for setting up our problem. The CMDP framework has been a common framework to study these problems and prior work in this area [1-5] has used this. The notion of safe exploration proposed by the reviewer is a very interesting one and we have added a new theoretical result to show that a variant of our method satisfies some guarantees on the proposed notion of safe exploration.
>
> **Added new result (Theorem 3)**: In addition, based on the reviewer’s suggestion, we have now added a new theoretical result (*Theorem 3*) that builds off of the existing Theorem 1, and shows that the cumulative number of failures incurred by our method grows sublinearly in the number of total amount of data collected, which indicates that failures go down at a “good” rate. Of course, as the reviewer points out, exploration with 0 failures is not possible unless the algorithm is provided with more information (for e.g., the functional form of safety constraint). Since we do not assume this, but are  able to show that the total number of failures grows sublinearly (Theorem 3), we show that our method minimizes the total number of failures at a good rate, while satisfying the constraints of the CMDP.
>
> **Removed “safety guarantee”**: To prevent the appearance of overclaiming that our method does “safe exploration”, we have updated the paper to remove the term “safety guarantee” and clearly define in the preliminaries, what the goal of the CMDP framework is -- the constraint violation probability should be bounded by the specified threshold for each intermediate policy. We have highlighted major modifications, wherever possible in blue in the paper. We next respond to detailed queries. We agree that there are different interpretations of "safe exploration" and we take a constrained RL view of safety [3,5], and define safe exploration in a CMDP as the process of ensuring the constraints of the CMDP are satisfied while exploring the environment to collect data samples.
>
> In the points below, we discuss the main concerns of the reviewer in bold, and provide our responses in plaintext.

---

> ### Author Response · Authors · 2020-11-20
> **Discussion**
>
> Kindly let us know if our response below addressed your concerns. We will be happy to answer if there are additional issues/questions.

---

### Decision · Program_Chairs · 2021-01-07
**Final Decision**

**Decision:**

Accept (Poster)

**Comment:**

Summary:
This paper introduces a different, interesting definition of safety in RL. The paper does a nice job of showing success with empirical results and providing bounds. I think it provides a nice contribution to the field.

Discussion:
The reviewers agree this paper should be accepted. The initial points brought up against the paper have been successfully addressed or mitigated.